# Comprehensive Elemental Profiling of Romanian Honey: Exploring Regional Variance, Honey Types, and Analyzed Metals for Sustainable Apicultural and Environmental Practices

**DOI:** 10.3390/foods13081253

**Published:** 2024-04-19

**Authors:** Florin Dumitru Bora, Andreea Flavia Andrecan, Anamaria Călugăr, Claudiu Ioan Bunea, Maria Popescu, Ioan Valentin Petrescu-Mag, Andrea Bunea

**Affiliations:** 1Viticulture and Oenology Department, Advanced Horticultural Research Institute of Transylvania, Faculty of Horticulture and Business in Rural Development, University of Agricultural Sciences and Veterinary Medicine Cluj-Napoca, 3-5 Mănăștur Street, 400372 Cluj-Napoca, Romania or florin-dumitru.bora@usamvcluj.ro (F.D.B.); claudiu.bunea@usamvcluj.ro (C.I.B.); 2Laboratory of Chromatography, Advanced Horticultural Research Institute of Transylvania, Faculty of Horticulture and Business for Rural Development, University of Agricultural Sciences and Veterinary Medicine, 400372 Cluj-Napoca, Romania; 3Fruit Growing and Pomology Department, Faculty of Horticulture and Business in Rural Development, University of Agricultural Sciences and Veterinary Medicine Cluj-Napoca, 3-5 Mănăștur Street, 400372 Cluj-Napoca, Romania; andreea.andrecan@usamvcluj.ro; 4Equine Clinic, Faculty of Veterinary Medicine, University of Agricultural Sciences and Veterinary Medicine Cluj-Napoca, 3-5 Mănăștur Street, 400372 Cluj-Napoca, Romania; maria.popescu@usamvcluj.ro; 5Department of Environmental Engineering and Protection, Faculty of Agriculture, University of Agricultural Sciences and Veterinary Medicine Cluj-Napoca, 3-5 Mănăștur Street, 400372 Cluj-Napoca, Romania; ioan.mag@usamvcluj.ro; 6Bioflux SRL, 54 Ceahlău Street, Cluj-Napoca, 400488 Cluj-Napoca, Romania; 7Doctoral School of Engineering, University of Oradea, 1 Universității Street, 410087 Oradea, Romania; 8Biochemistry Department, Faculty of Animal Science and Biotechnology, University of Agricultural Sciences and Veterinary Medicine Cluj-Napoca, 3-5 Mănăștur Street, 400372 Cluj-Napoca, Romania; andrea.bunea@usamvcluj.ro

**Keywords:** elemental correlation, environmental impact, geochemical fingerprint, multivariate analysis, regional differentiation

## Abstract

We investigated the mineral concentrations of 61 honey samples from eight Romanian regions, employing advanced techniques to assess 30 chemical elements. Potassium emerged as the dominant element, showcasing significant variations across geographical locations. Essential minerals like calcium, magnesium, sodium, and manganese maintained consistent levels, while zinc, copper, and chromium were present in smaller proportions. Critically, lead and cadmium levels exceeded established safety limits in some samples, suggesting potential environmental contamination. Additionally, elevated levels of lithium, strontium, nickel, and aluminum were detected, hinting at possible atmospheric pollution. These findings highlight the importance of regional analysis, as mineral content varied significantly between locations. Furthermore, correlation analysis revealed interdependencies among elements, suggesting shared environmental influences. Advanced statistical techniques like hierarchical clustering and principal component analysis effectively captured the impact of geographical origin on honey composition. These insights contribute valuable information for future efforts in honey quality control, traceability systems, and regulatory measures. By providing valuable insights into environmental influences on honey composition, this study informs future research endeavors and paves the way for the development of robust regulatory measures to ensure honey safety for consumers.

## 1. Introduction

Undoubtedly, honey stands out as the most crucial bee product, and, historically, it was the first bee-derived substance employed by humans in ancient times [1]. The honey created by bees is a concentrated natural sugar solution, comprising a complex blend of carbohydrates. It is consumed as a food with significant nutritional and health benefits [2]. Honey, primarily consumed by children over one year old, adults, and individuals with weakened health, is not only a food product: it is also used for healing purposes [3]. Ensuring its quality is crucial, necessitating an assessment of food safety and environmental considerations. The study of honey quality is paramount, aiming to guarantee its purity, absence of harmful components, and minimal levels of substances like heavy metals, emphasizing its significance both as a safe food product and an environmental biomonitor [1].

International standards set by organizations like the Codex Alimentarius (CA, 2010) and the European Community (EU) (EU Council, 2002) define honey as a natural sweet substance produced by honey bees. It originates from the nectar of plants or secretions from living plant parts and can also include honeydew, a sugary excretion left by plant-sucking insects on plants. Honey bees collect these materials, transform them with their enzymes, and then deposit, dehydrate, store, and leave them in honeycombs to mature. During this process, they also add their own specific substances [4]. According to these standards, commercial honey must not contain any added food ingredients, including food additives, and no other substances should be introduced aside from honey. The honey should be free from objectionable matter, undesirable flavors, aromas, or taints acquired during processing and storage. It must not have begun fermentation or effervescence. Removal of pollen or any constituent inherent to honey is only permitted if necessary for eliminating foreign inorganic or organic materials [4]. Furthermore, honey should not undergo excessive heating or processing that alters its essential composition and impairs its quality. Consequently, adherence to these criteria ensures the authenticity of honey [4,5].

Honey, a natural and nutritious food, is a bee-made treasure crafted from nectar (flower secretions) and honeydew (plant sap). It may also contain flecks of pollen, hinting at the floral sources the bees visited [4]. The composition of honey primarily comprises a blend of carbohydrates, with fructose and glucose making up 85–95% of the total sugars; the ratios of these sugars in honey govern its overall granulation process, with a higher fructose concentration resulting in a longer liquid state due to the lower solubility of glucose [2,6,7]. The remaining carbohydrates in honey consist of two or more bonded fructose and glucose molecules, with only a trace of polysaccharide residues present [8]. In addition to its primary sugar content, honey contains a variety of minor components, including fat-soluble vitamins, proteins, amino acids, antioxidant-rich flavonoids, organic acids, enzymes, minerals, and hydroxymethylfurfural, the last of which acts as an indicator of honey’s freshness [8,9]. From a commercial standpoint, honey can be classified as either monofloral or polyfloral based on melissopalynological analysis. Monofloral honey, derived primarily from a single pollen source, holds greater economic value due to its higher concentration of single-origin pollen, accounting for up to 45% of the solid residue [7]. Monofloral honey is derived from nectar predominantly sourced from a single plant species [10], offering distinct sensory attributes and commanding a premium price; this is in contrast to polyfloral honey, which results from the blending of various nectar sources by honey bees during the honey-making process using stored nectar and lacks the specific characteristics of monofloral varieties [10,11]. The unique characteristics and composition of honey are directly influenced by the botanical sources of the nectar and the bees’ own secretions [12]. Apart from its diverse nutritional and medicinal attributes, honey possesses preventive (prophylactic) qualities attributed to its wide array of chemical constituents [12]. To achieve the desired therapeutic benefits, honey must be devoid of contaminants present due to its widespread use as a natural and pleasant sweetener, which has sparked growing interest in honey research [13]. Research underscores that honeybees, in their foraging activities covering vast areas exceeding 7 km^2^, maintain continuous contact with air, soil, water, leaves, and branches [12].

The mineral content in honey is directly impacted by the chemical makeup of nectar sourced from diverse plant species [14]. Honey’s chemical composition is not solely determined by the botanical sources of nectar and bee secretions; it is also influenced by a complex interplay of factors, including ecological and soil conditions, bee species, honey maturation, colony health, ecoclimatic variables, and seasonal fluctuations. These factors collectively contribute to honey’s unique properties and diverse characteristics [14]. Research suggests that darker honey varieties, such as amber honey, generally contain higher concentrations of macroelements (calcium, magnesium, potassium, and sodium), microelements (copper, iron, manganese, and zinc), and trace elements (aluminum, cadmium, and nickel) compared to lighter honey varieties [15]. In addition to the natural factors influencing honey’s mineral content, anthropogenic influences, such as emissions from major roads, urban areas, and industrial zones, can also lead to elevated concentrations of certain elements in honey, including aluminum (Al), barium (Ba), cadmium (Cd), copper (Cu), manganese (Mn), nickel (Ni), lead (Pb), and zinc (Zn) [14,16]. To date, 31 elements have been detected in honey from various botanical sources [17], with macroelements and microelements found within a narrower concentration range, typically ranging from 0.04% to 0.2% [18]. Studies have demonstrated that the overall mineral content in honey can surpass 1%, with all constituents being dissolved in the water contained within it (with a content ranging from 13.4% to 22.9%), contributing to the honey’s unique color and texture, whether it is in a liquid or semi-liquid state [16]. Metals primarily move from the soil into honey plants through root systems, subsequently transferring into the nectar and eventually becoming part of the honey produced by bees [19]. Hence, the chemical makeup of honey, particularly concerning major and minor metals like Ca, K, Mg, Mn, and Na, is impacted by the soil’s composition, which is influenced by factors such as geographical origin, regional conditions, climatic changes in bees’ foraging area, and volcanic and hydrothermal activity [16].

Honey mislabeling and adulteration are significant global concerns. Adulteration practices include diluting honey with water, adding sugars or syrups (like corn syrup), feeding bees sugar instead of nectar, and introducing artificial honey. Additionally, misrepresenting the floral or geographical origin of honey is a common problem. Ensuring food safety and quality control requires robust food authentication methods. Regulatory bodies, food processors, retailers, and consumers all have a vested interest in verifying the authenticity and quality of honey. Unfortunately, incidents of honey mislabeling and adulteration continue to occur [4,20,21,22,23].

The chemical analysis of honey reveals valuable insights into its geographical and botanical origins [4]. Researchers use fingerprinting techniques to study volatile chemicals, providing details on variations tied to floral origins and honey processing [24]. Saccharides are scrutinized to detect adulteration, while enzyme activities, fermentation products, and analyses of minerals offer clues about botanical and geographical origins [25]. Although some methods face limitations in accurately distinguishing geographical origins, the proteins in honey emerge as a promising tool for identifying both botanical and geographical sources [26].

Honey produced in proximity to industrial activities, such as mining, smelting, and urban processes, often exhibits elevated levels of heavy metals, including arsenic (As), cadmium (Cd), chromium (Cr), mercury (Hg), nickel (Ni), and lead (Pb). The presence of heavy metals can also be attributed to agrochemical usage, such as arsenic-containing pesticides or fertilizers with organic mercury or cadmium. This heavy metal pollution poses a significant concern, as the deficiency, excess, or imbalance of microelements in honey can lead to health issues in humans [27,28]. Microelements play a crucial role in biological accumulation, actively contributing to natural physiological development, metabolism, and overall metabolic processes. Essential mineral elements like Na, K, Ca, Fe, Zn, Cu, and Mn are integral for the biological metabolism of living organisms [1]. Elements like Pb, Cd, Hg, and As are categorized as environmental micropollutants, being toxic or non-essential for living organisms. Elevated concentrations of these trace elements can prove lethal, as the body lacks the capacity to metabolize heavy metals efficiently [1]. Heavy metal contamination in honey can lead to a range of health problems, including headaches, metabolic disorders, respiratory issues, nausea, vomiting, and damage to the brain, kidneys, nervous system, and red blood cells [29].

Romania has significant beekeeping potential and extensive experience in the field, attributed to favorable climatic conditions and diverse melliferous resources [30]. Acacia, linden, rape, sunflower, and polyfloral and honeydew honeys are the most produced types in this country. Additionally, Romania annually produces small quantities of rare honeys like mint, raspberry, and lavender [31]. In 2010, Romania ranked third in honey production and fourth in the number of bee families in the European Union [30]. By the end of 2015, Romania had achieved the highest honey production level in the EU, reaching 35,000 tons compared to 20,000 tons in 2014, surpassing Spain and Hungary [30]. This significant increase was attributed to the utilization of funds from the National Bee Programme (2008–2010), aimed at expanding the bee population and promoting beekeeping [30].

Several scientific studies are summarized in Appendix A. These studies detail the concentration levels of elements of interest in honey. For routine analysis in laboratories, the common methods for determining element levels in honey samples include atomic absorption spectroscopy (AAS) techniques, like flame AAS (FAAS), as well as inductively coupled plasma (ICP) methods such as ICP-MS, ICP-OES, and ICP-AES. Microwave plasma atomic emission spectrometry (MP-AES) and graphite furnace atomic absorption spectrometry (GFAAS) are also employed [32,33,34,35,36,37,38,39,40,41,42,43,44].

This study was undertaken with the objective of assessing the concentrations of 30 chemical elements categorized into macroelements (^19^K, ^23^Na, ^24^Mg, and ^43^Ca), microelements [trace elements (^7^L, ^27^Al, ^56^Fe, ^64^Cu, ^65^Zn, and ^88^Sr) and ultra-trace elements (^9^Be, ^51^V, ^52^Cr, ^55^Mn, ^59^Co, ^60^Ni, ^70^Ga, ^79^Se, ^85^Rb, ^204^Tl, ^208^Ag, ^209^Bi, ^115^In, ^113^Cs, and ^137^Ba)] and heavy metals (^75^As, ^111^Cd, ^201^Hg, ^208^Pb, and ^238^U) throughout a dataset comprising 61 honey samples collected from the entire expanse of Romania. These samples originated from 8 (the southeast (22 samples), east (7), northeast (3), center (5), west (2), south (3), southwest (5) and southeast (14)) distinct regions characterized by varying geographical and environmental attributes. The analysis was conducted utilizing inductively coupled plasma-mass emission spectrometry (ICP-MS) techniques. The principal aims addressed by the present investigation encompass the following: *(i) The elucidation of geographical and botanical influences on honey composition*, involving the investigation of the elemental compositions of honey from 61 samples across different regions in Romania, focusing on major, minor, trace, and heavy metal elements. These findings were then correlated with geographical and botanical factors to unravel the complex dynamics influencing honey composition. *(ii) The assessment of heavy metal contamination and health risks:* This study aimed to evaluate heavy metal concentrations in honey samples, with a particular focus on lead (Pb) and cadmium (Cd) alongside other potentially concerning elements. We analyzed samples from various regions to identify areas where heavy metal levels exceed established legal limits. Furthermore, a comprehensive health risk assessment was conducted based on the detected heavy metal concentrations. This assessment will provide valuable insights into the potential health risks associated with consuming different honey types. *(iii) Exploring correlations and interdependencies in honey composition:* We conducted correlation analysis, hierarchical clustering, and principal component analysis to unveil interrelationships among elements, highlighting shared environmental influences and contamination pathways. We explore the significance of positive and negative correlations in predicting mineral behavior and identifying potential sources of contamination. *(iv) Region-specific quality control measures:* We examine regional disparities in mineral content and heavy metal concentrations, emphasizing the importance of region-specific monitoring and regulation for honey quality control. To safeguard the safety and authenticity of honey products across Romania’s diverse regions, targeted quality control measures and regulatory interventions are essential. Implementing stricter monitoring programs and enhancing laboratory testing capabilities for heavy metal and adulteration detection are crucial steps. Additionally, establishing clear labeling requirements and traceability systems can effectively combat misrepresentation and ensure consumers receive genuine honey products. Furthermore, supporting beekeeping practices that promote sustainable bee health and minimize environmental contamination is paramount for long-term honey quality assurance.

## 2. Materials and Methods

### 2.1. Research Location

A comprehensive collection comprising 61 honey samples from *Apis mellifera*, designated as H1–H33, was assembled during the years 2018 and 2021. These samples were obtained from eight geographically distinct regions within Romania: southeast (22 samples), east (7), northeast (3), center (5), west (2), south (3), southwest (5), and southeast (14). These provinces encompass the following regions: southeast (Galați (6), Târgu Bujor (2), Tecuci (3), and Brăila (7)), east (Vaslui (7)), northeast (Satu Mare (3)), center (Sibiu (5)), west (Arad (2)), south (Teleorman (3)), southwest (Râmnicu Vâlcea (2) and Mehedinți (3)), and southeast (Botoșani (8) and Iași (6)). The honey samples consisted of multifloral (n = 11), linden (n = 9), acacia (n = 11), sunflower (n = 16), spring rape (n = 2), autumn rape (n = 1), lavender (n = 3), acacia + linden (n = 4), chestnut (n = 2), and honeydew (n = 2). A cartographic depiction of the provenance of the acquired honey samples is delineated in Appendix A.

### 2.2. Collection of Honey Samples

#### Honey Sampling

The samples, sourced from the local beekeepers’ association (with guaranteed origins), were produced using traditional methods in the respective honey-producing regions. Each sample was collected in clean, sealed glass jars, maintaining unpasteurized status. Following collection, honey samples were promptly transported to a laboratory and stored in glass bottles under refrigeration (4–5 °C) in a dark environment until analysis. This meticulous handling procedure was designed to preserve the integrity and representativeness of the samples for subsequent laboratory testing. All procedures adhered to hygienic practices outlined by the Codex Alimentarius Commission (2003), with clean facilities and equipment. The sample set included 15 multifloral and 46 monofloral honeys. The quantity of samples within each floral classification was indicative of the typology of the honeys generated in the investigated region, reflecting the unique conditions prevalent during the respective years of sample collection. The acquisition of honey samples was executed with meticulous attention to ensuring their representativeness for the designated honey lot. Specifically, the samples comprised honey sourced from three distinct beehives within the same production lot. The extraction process involved centrifugation, and the resultant honey was carefully collected in glass jars. These stored samples were preserved until they were required for subsequent analysis. The treatment and preparation of the samples adhered rigorously to the guidelines delineated in the Harmonized Methods of the International Honey Commission [45]. In Appendix A, we have summarized the descriptive characteristics of honey samples, encompassing sample codes, honey specifics, classification, geographical origins, harvest years, extraction methods, bee species, environmental factors, and anthropogenic influences.

### 2.3. Sample Preparation and Microwave Digestion Procedure

#### 2.3.1. Analytical Approaches for Heavy Metal Analysis

To facilitate sample handling and ensure consistency throughout analysis, honey samples were gently warmed in a water bath at 65 °C until they liquefied. Ideally, samples were heated in their original sealed containers within a rotating water bath for 30 min to maintain homogeneity [46,47]. Honey samples were gently stirred while being warmed in the water bath to ensure even distribution (homogeneity) throughout the sample. Since honey dissolves readily in water, and metals can be directly measured in liquid solutions, honey samples are typically dissolved in water or a weak acid solution before analysis [16,48]. Following the initial sample preparation (involving ashing or digestion), the remaining residues are redissolved in either an acidic solution or ultrapure water. This step helps extract the metal components by breaking down the honey’s organic structure and allowing the metals to transfer more efficiently into the solution for analysis [48]. To ensure consistent analysis, all honey samples were homogenized. For liquid honey samples (chestnut, acacia, and polyfloral), this involved vigorous shaking. Crystallized honey samples (seven in total) were gently warmed in a water bath at 65 °C for 30 min to dissolve the crystals and achieve a uniform consistency. Using a precise balance, exactly 1 g of each homogenized honey sample was carefully measured and transferred into polypropylene tubes. To prepare for analysis, each honey sample (1 g) was dissolved in 20 mL of high-purity deionized water (specific resistivity: 18.2 MΩ × cm^−1^) using the Milli-Q Integral Ultrapure Water-Type 1 system. The solution was then heated to 65 °C to facilitate complete dissolution [48].

#### 2.3.2. The Microwave-Assisted Digestion Methodology

Alternatively, instead of water dissolution, some honey samples may undergo a mineralization process using a Milestone START D Microwave Digestion System (Sorisole, Italy). For this method, approximately 0.5 g of the sample is carefully weighed and placed into a clean Teflon digestion vessel. Next, 12 mL of aqua regia, a strong acid mixture of 9 mL of hydrochloric acid (HCl) and 3 mL of nitric acid (HNO_3_), is added. Following a 15-min incubation, the system utilizes microwave technology to complete the mineralization process [48]. For detailed information on the operational parameters of the Milestone START D Microwave Digestion System (Sorisole, Italy), please refer to Appendix A.

### 2.4. Basic ICP-MS Analytical Instrumental Parameters

The elemental compositions of honey samples were determined using inductively coupled plasma mass spectrometry (ICP-MS) (iCAP Q ICP-MS, Thermo Fisher Scientific, Waltham, MA, USA). This technique allowed us to quantify a broad range of elements, including macroelements (e.g., potassium, sodium, magnesium, and calcium), microelements (e.g., aluminum, iron, copper, and zinc), ultra-trace elements (e.g., beryllium, vanadium, and chromium), and heavy metals (e.g., arsenic, cadmium, mercury, lead, and uranium). The ICP-MS system was equipped with an autosampler, a nebulizer, and a collision cell for mitigating common interferences. Samples were placed into the instrument using a nebulizer connected to a spray chamber, employing a standard ICP-MS torch. This analytical approach enables highly sensitive detection for a comprehensive assessment of the elemental profiles of honey. Before analyzing the samples, the ICP-MS system underwent a 45 min equilibration period to ensure stable operating conditions. During this time, a mass calibration verification was performed using a standard solution containing various elements. An automated tuning process then optimized the instrument for sensitivity and minimal interference from background noise. To ensure optimal performance throughout the analysis, the ICP-MS system was calibrated daily. This included maximizing sensitivity for target elements (M+ ions) while monitoring for minimal interference from doubly charged ions and oxides. High-purity argon (Ar 5.0, 99.999%) and helium (He 6.0, 99.9999%) gases (Messer, Gumpoldskirchen, Austria) were used. Each honey sample was analyzed twice, with each analysis consisting of seven repeated measurements (replicates) to allow for improved accuracy. For comprehensive details on the ICP-MS operating parameters, please refer to Appendix A. It is important to note that the optimization of general ICP-MS instrumental settings is a well-established procedure documented in previous research [48].

### 2.5. Chemical and Apparatus

Analytical-grade chemicals and reagents were obtained from trusted suppliers (Merck and Sigma Aldrich, Darmstadt, Germany) to ensure accuracy. Ultra-pure nitric acid (65%, trace-metal analysis grade) and hydrogen peroxide (30%, trace-metal analysis grade) were used for sample preparation. External calibration with proper dilution was the chosen method for analysis. Additionally, internal standards (Ge, Tb, Rh, and Sc in 1% ultra-pure nitric acid) were added to all samples, blanks, and standards at a concentration of 50 µg/L to account for potential instrument variations. A high-purity ICP Multi-Element Standard Solution (Merck, Darmstadt, Germany) was used for calibration curve construction. The specific calibration and internal standard addition procedures used were established in previous research [48]. To ensure minimal contamination, high-purity deionized water (specific resistivity: 18.2 MΩ × cm^−1^) obtained from a Milli-Q system was used for preparing all aqueous solutions, including standards and sample dilutions. Teflon digestion vessels were meticulously cleaned with nitric acid before each mineralization process, which was performed in triplicate for both soil and honey samples. The digestion system itself could hold up to six vessels (five for samples and one blank) made from a special Teflon material (TFM-PTEE). All flasks used throughout the experiment underwent a rigorous cleaning process involving a 24 h rinse with concentrated nitric acid followed by multiple washes with deionized water. A high-precision balance (KERN ADB 100-4) was employed for the accurate weighing of honey samples and preparation of calibration and working solutions.

### 2.6. Chemical Analysis Quality Control

The limits of detection (LoDs) and quantification (LoQs) for inorganic arsenic, lead, and polycyclic aromatic hydrocarbons (PAHs) were established following the analytical performance criteria outlined in Commission Regulation (EU) No. 2016/582 (amending Regulation (EC) No. 333/2007) [48]. Limits of Detection (LoDs) and Quantification (LoQs) for the analyzed elements were determined based on the standard deviation (σ) of blank measurements. Specifically, a value of 3σ was used for LoDs, and 10σ was used for LoQs (detailed in Appendix A). Repeatability of the analysis was evaluated using the Horwitz ratio (HorRat). This ratio is calculated by dividing the measured relative standard deviation (RSDr) of samples by the theoretical RSDs predicted using the Horwitz equation [48]. For reliable analysis, the calculated Limits of Detection (LoDs) and Limits of Quantification (LoQs) needed to be below a threshold value of 2 (refer to Appendix A for detailed results). Additional validation parameters like precision, accuracy, recovery, and uncertainty are presented in Appendix A. Calibration standards were prepared at various concentrations (2.5, 5, 10, 25, and 50 μL) from a high-purity ICP multi-element standard solution. Accuracy was evaluated by spiking a known amount of the target metal into a sample aliquot, followed by analysis alongside the original sample. Repeatability of the analysis was expressed as the relative standard deviation (RSD%) calculated from triplicate analyses. Recovery tests were also performed using honey samples spiked with a specific concentration (5 μL) of the analyte. The average recovery across three replicates (n = 3) ranged from 90.32% to 113.12%, with uncertainties between 9% and 23%.

### 2.7. Statistical Analysis

Data analysis was performed using a combination of software tools. Microsoft Excel 365 and Addinsoft (2018) were used for initial data exploration, calculating descriptive statistics (averages, medians, standard deviations, and correlations), and exploratory analyses (hierarchical clustering and principal component analysis). Data precision was assessed by calculating standard deviations (SD). All data were reported as means ± SD. Statistical analysis for hypothesis testing was then conducted using SPSS Version 24. Data were presented as means (averaged from three replicates) with standard deviations. A two-way analysis of variance (ANOVA) was performed in SPSS to investigate the effects of various factors on the concentrations of heavy metals in both soil and honey samples. Following the acquisition of significant results from the ANOVA, further analysis using Duncan’s test (*p* ≤ 0.005) was used to identify specific differences between groups.

## 3. Results and Discussion

### 3.1. Analyzing Elemental Distribution in Various Honey Types—Mineral Profiles

A total of 61 honey samples were subjected to comprehensive analysis to ascertain the presence of major, minor, trace, and heavy metal elements. It was observed that none of the honey samples exhibited detectable concentrations of beryllium (Be), vanadium (V), cobalt (Co), gallium (Ga), selenium (Se), rubidium (Rb), telluride (Tl), silver (Ag), bismuth (Bi), indium (In), cesium (Cs), barium (Ba), arsenic (As), mercury (Hg), or uranium (U). Consequently, these specific elements, lacking a measurable presence in the honey samples, are omitted from further discussion in this analytical context. The study findings are systematically organized and articulated in Table 1, Table 2 and Table 3, wherein a comprehensive breakdown is provided based on distinct criteria, namely, the categorization of honey types, production regions, and production years. Each table encapsulates a nuanced representation of the research outcomes, allowing for a detailed examination of the variations and trends for different types of honey, diverse production areas, and the temporal dimension of production years. This meticulous presentation serves to enhance the interpretability and accessibility of the research data, facilitating a more in-depth analysis of the interrelationships and patterns that emerge from the investigated parameters.

Potassium (K) emerged as the predominant element, constituting 84.04% (Appendix A) of the elemental composition, exhibiting an average concentration of 809.41 ± 577.92 mg/kg and displaying a considerable variability within the range of 236.08–2116.41 mg/kg. K contributes substantively, accounting for approximately one-third of the overall mineral composition within honey. Its presence, combined with that of other significant inorganic constituents, imparts crucial insights into the nutritional attributes of this product [44]. Analyzing the distribution of K within the assessed honey samples revealed that sunflower honey exhibited the lowest (45.35 ± 30.40 mg/kg) concentration of this element, while chestnut honey registered the highest (2116.41 ± 183.26 mg/kg) concentration. The influence of geographical origin on honey samples is evident in the observed distribution of potassium (K). Specifically, it is evident that honey derived from rape in Tecuci exhibited the highest recorded values (1012.05 ± 20.89 mg/kg—autumn rape and 1284.66 ± 52.33 mg/kg—spring rape). The concentration of K in bee products exhibits a robust correlation with the geographical attributes of the pollen and nectar production region, including soil characteristics and agricultural practices [16]. Chudzinska et al. (2011) [49] additionally validated the utility of potassium (K) as a botanical marker in cases where the authenticity of a sample is under scrutiny.

The observed concentrations of potassium (K) align closely with the documented findings in the study conducted by Pătruică et al. (2022) [36]. In the Banat region of Romania, the potassium (K) concentrations were found to be 65.089 mg/kg in sunflower honey and 85.706 mg/kg in linden honey (Appendix A). A comparable correspondence was noted with respect to the outcomes detailed in the research conducted by Purcarea et al. (2017) [37], who examined honey samples sourced from heather (*Calluna vulgaris*) and acacia in the Bihor area of Romania. In their study, K concentrations of 1680.685 mg/kg and 213.552 mg/kg, respectively, were recorded (Appendix A). The findings indicated that potassium (K) constituted the most prevalent major element in honey, with an average concentration of 809.41 ± 577.92 mg/kg. This mean value was comparatively lower than the values documented in honey samples originating from Hungary (2069.1 mg/kg—honeydew; 1892.7 mg/kg—forest; and 2466.6 mg/kg—chestnut; Sajtos et al. 2019 [44]). Conversely, in comparison to the results obtained for honey samples collected in Baia Mare, Romania (non-detectable—polyfloral; Berinde et al. (2023) [38]); France (non-detectable—polyfloral; Devillers et al. (2002) [41]), and Greece (non-detectable—fir; Louppis et al. (2017) [42]), the concentrations recorded were significantly higher. Regarding the results derived from honey samples from Bulgaria (126–1628 mg/kg—unifloral; Atanassova et al. (2012) [40]), Algeria (808.00 mg/kg—lavender; 460.00 mg/kg—rosemary; 418.00 mg/kg—multifloral; Bereksi-Reguig et al. (2022) [43]) and Hungary (327.9 mg/kg—acacia; 696.5 mg/kg—multifloral; Sajtos et al. 2019 [44]), the results are comparable. The second-most-prevalent element, Ca, constituted 8.05% (Appendix A) of the elemental composition, with an average concentration of 77.49 ± 48.23 mg/kg, displaying considerable variability within the range of 14.87–303.13 mg/kg. Minimum and maximum values were observed in multifloral honey from Botoșani and Râmnicu Vâlcea, respectively. Simultaneous recording of extreme values, both minimum and maximum, for a specific honey attribute indicates significant fluctuations in its quality or composition at a particular moment. The potential contributions of various conditions, such as weather or nectar sources, to this variability cannot be disregarded. A meticulous assessment of the context and influencing factors is crucial for attaining a deeper understanding of the origins of these fluctuations. Notably, changes in the surrounding environment or the bees’ food sources during honey collection may have led to a diverse array of characteristics in the honey. The analysis of this element in relation to the sample origins reveals that honey from Târgu Bujoru exhibited the highest concentration, amounting to 195.53 mg/kg, while that from Râmnicu Vâlcea showed the second-highest concentration, 162.71 mg/kg. Upon comparison with the findings from this study, it is evident that the calcium concentration aligns consistently with that from previous research on honey sourced from Romania, specifically Domasnea (185.80 ppm—acacia), Farling (197.10 ppm—acacia), and Bala (195.05 ppm—acacia), conducted by Pătruică et al. (2008) [32]; Romania, specifically Dolj, Mehedinți, and Gorj (5.8–76.46 ppm—acacia; lime sunflower and polyfloral honey), conducted by Pătruică et al. (2009) [33]; and Romania, specifically Banat (32.521 mg/kg—knotweed, 70.547 mg/kg—linden, 37.370 mg/kg—acacia 54.280 mg/kg—sunflower), conducted by Purcarea et al. (2017) [37]. These findings are in line with the outcomes observed in analyses of honey samples conducted worldwide (Bulgaria—unifloral, Atanassova et al. (2012) [40]; France—polyfloral, Devillers et al. (2002) [41]; and Hungary—sunflower, Sajtos et al. 2019 [44]. The third-most-prevalent elements, Mg, Na, and Fe, make up 3.86%, 2.16%, and 1.52% (Appendix A) of the elemental composition, with average concentrations of 37.14 ± 22.39 mg/kg (Mg), 20.76 ± 17.20 mg/kg (Na), and 14.60 ± 20.72 mg/kg (Fe). Lavender honey samples from the Vaslui region exhibited the highest concentrations of Mg (84.45 ± 7.75 mg/kg) and Fe (75.91 ± 5.98 mg/kg), while honeydew exhibited the highest Na values (69.42 ± 6.32 mg/kg). Conversely, the lowest values were documented for chestnut, acacia, and linden honey samples from Botoșani, Iași, and Râmnicu Vâlcea. Regarding the obtained concentrations of Na and Mg, they align with findings from other studies on honey samples sourced from Romania, e.g., the studies by Pătruică et al. (2008) [32] regarding Na in Domasnea (26.062 ppm—acacia), Farling (28.080 ppm—acacia), and Bala (27.165 ppm—acacia) and again by Pătruică et al. (2022) [36] regarding Mg in Banat ((35.280 mg/kg—knotweed), (40.700 mg/kg—linden), (35.179 mg/kg—acacia), and (38.097 mg/kg—sunflower)). Concerning Fe concentrations, the obtained results indicate significantly lower values compared to those reported by Ciobanu et al. (2016) [35] for Timiș (80.32 ppm—linden 1, 67.89 ppm—linden 2, 47.24 ppm—rape, and 23.18 ppm—acacia). Also, in the reporting of research findings on honey samples from Bulgaria [40], France [41], Greece [42], and Hungary [44], the findings of this study align with the results of previous research. Mn exhibited its highest values in honey samples from Botoșani (7.17 ± 2.81 mg/kg—linden (2022)), Brăila (5.42 ± 0.42 mg/kg—linden), and Arad (4.39 ± 1.34 mg/kg—acacia). Twelve honey samples, constituting 16.67% of the total analyzed, exhibited manganese concentrations below the detection limit (LoQ for ^55^Mn 0.039 µ/L). The honey samples with manganese concentrations below the detection limit were sourced from Tulcea and Mehedinți and, for a portion of the samples, from Vaslui, Satu Mare, Iași, and Râmnicu Vâlcea. Similar to the findings for the previously discussed elements, the obtained results are consistent with analyses conducted on honey samples from Romania [35,36,37] or across the globe [41,42,43,44].

Remarkably, the concentrations of potassium (K), calcium (Ca), phosphorus (P), magnesium (Mg), sodium (Na), and manganese (Mn) were consistently high and evenly distributed across all the honey samples examined. These elements in the honey samples are postulated to have originated from natural or lithogenic sources, primarily linked to soil composition. However, it is plausible that fertilizers may have contributed to these concentrations, with a particular emphasis on potassium and phosphorus. Additionally, the concentration of manganese could be subject to the influence of anthropogenic factors [1].

The elements Zn, Cu, and Cr contribute a minor proportion to the chemical composition of honey, constituting 0.07% (Zn) and 0.06% of the elemental composition for Cu and Cr, respectively (Appendix A). Their average concentrations are 0.71 ± 0.68 mg/kg for Zn, 0.55 ± 0.68 mg/kg for Cu, and 0.54 ± 0.53 mg/kg for Cr. The highest zinc concentrations were observed in multifloral honey from the Râmnicu Vâlcea region (6.46 ± 1.41 mg/kg), as well as sunflower honey from Botoșani (3.43 ± 0.27 mg/kg), Vaslui (3.23 ± 0.33 mg/kg), and the Târgu Bujor area (2.63 ± 0.49 mg/kg). Concerning the recorded Cu concentrations, a notable proportion of the samples exhibited elevated values of this element. Heightened concentrations were observed in multifloral honey samples (0.64 ± 0.44 mg/kg), as well as in lavender (2.40 ± 1.25 mg/kg), sunflower (1.83 ± 0.55 mg/kg), acacia (2.01 ± 1.71 mg/kg), multifloral (1.71 ± 1.84 mg/kg), and sunflower (4.31 ± 0.39 mg/kg) samples obtained in the Galați, Vaslui, Satu Mare, Brăila, and Tulcea regions. The values for Zn and Cu in the analyzed samples surpass the legally permitted maximum concentrations. In contrast to the Zn and Cu concentrations in the honey samples, Cr exhibited the highest values in samples from Sibiu (3.29 ± 0.57 mg/kg—acacia), Iași (2.34 ± 0.71 mg/kg—acacia + linden), Vaslui (1.74 ± 0.73 mg/kg—linden), and Mehedinți (3.66 ± 1.63 mg/kg—acacia; 1.15 ± 0.18 mg/kg—sunflower).

The possible reasons for high Zn and Cu levels in honey that exceed the legal limits include environmental contamination (plants absorbing metals from their surroundings), agricultural practices (the use of metal-containing fertilizers or pesticides affecting bees and honey), regional differences (practices in local industries, mining, or agriculture that elevate metal content), floral sources (certain flowers accumulating more metals), and anthropogenic factors (industrial emissions or improper waste disposal introducing additional metals). A detailed analysis specific to the honey collection regions is crucial for identifying the precise causes. Comparing the findings of this study with national [32,33,34,35,36,37,38] and international [39,40,41,42,43,44] research reveals challenges concerning the concentrations of Zn and Cu in honey from Romania [32,33,34,35,36,37,38] as well as from Italy [39], Bulgaria [40], France [41], and Hungary [44]. Elevated concentrations of Zn and Cu in honey, both from Romania and other countries, may raise concerns about food safety. Possible explanations include environmental pollution, agricultural practices involving metal-containing pesticides and fertilizers, and geographic variability influencing metal content in honey. Monitoring and careful regulation are essential to ensure honey quality and consumer safety. The findings related to Cr align with results for both national and international honey samples.

The following significant elements, Pb and Cd, contribute 0.09% and 0.03% (Appendix A), respectively, to the elemental composition of honey, with average concentrations of 0.09 ± 0.07 mg/kg (Pb) and 0.03 ± 0.01 mg/kg (Cd). Pb exhibits elevated concentrations in honey samples from the Brăila, Tulcea, Botoșani, Iași, and Mehedinți regions. The highest values were documented in linden honey from Botoșani (0.31 ± 0.29 mg/kg), with sunflower honey from Iași also showing notable results (0.27 ± 0.05 mg/kg). The results indicate that honey samples from the Tulcea and Botoșani regions exhibited the highest concentrations of Cd. In both regions, multiflora honey demonstrated elevated concentrations (0.17 ± 0.04 mg/kg—Tulcea; 0.13 ± 0.14 mg/kg—Botoșani). Additionally, high concentrations were observed in sunflower honey (0.12 ± 0.01 mg/kg—Tulcea) and honeydew honey (0.13 ± 0.14 mg/kg—Botoșani). Conversely, the honey samples from Tecuci (Pb and Cd), Braila (Cd), Satu Mare (Pb), Sibiu (Pb and Cd), and Râmnicu Vâlcea (Cd) exhibited concentrations of Pb and Cd below the detection limit. A considerable portion of the analyzed honey samples surpass the legal maximum limits for Cd and Pb concentrations. It is noteworthy that Cd and Pb, both heavy metals with high concentrations, are prevalent in similar honey production areas. Elevated levels may result from environmental contamination, agricultural practices, regional variations, floral sources, and anthropogenic factors. Varied regional levels indicate potential contributions from local industrial activities, mining, or specific agricultural practices, underscoring the importance of detailed analyses. The obtained results align with findings from previous studies, such as with respect to Cd concentrations ranging from 0.05 to 3.81 mg/kg in polyfloral honey from Copșa Mică, Romania [34]; 0.130 mg/kg in knotweed honey from Banat, Romania [36]; and ND-0.78 mg/kg in polyfloral honey from Baia Mare, Romania [38]. The obtained Pb results align with the findings of Bartha et al. (2020) [34] (0.76–3.41 mg/kg), Pătruică et al. (2022) [36] (0.163 mg/kg), and Berinde et al. (2013) [38] (0.12–20.34 mg/kg), respectively. The reported results confirm the presence of heavy metal pollution in the investigated areas, with said pollution being particularly evident in honey samples from Copşa Mica and Baia Mare. These areas were distinctly identified as pollution zones in the conducted investigations. A potential next step in this research is to identify the sources of pollution in regions where elevated Pb and Cd values are observed.

The highest concentrations of Li, Sr, Ni, and Al were documented in honey samples from Tulcea, Botoșani, Iași, and Sibiu regions. In terms of their proportions relative to the total honey concentration, Li registered 0.02%, while Sr, Ni, and Al corresponded to 0.01% (Appendix A). Li and Sr exhibited average values of 0.19 ± 0.17 mg/kg and 0.05 ± 0.04 mg/kg, respectively. The highest concentrations were observed in acacia honey regarding Li (0.81 ± 0.32 mg/kg) and sunflower honey regarding Sr (0.87 ± 0.27 µg/kg). In contrast, the peak concentrations of Ni were recorded in acacia honey from Iasi (0.29 ± 0.06 mg/kg) and Sibiu (0.30 ± 0.09 mg/kg). Among the honey samples analyzed, chestnut honey from Satu Mare exhibited the highest aluminum (Al) concentration, 0.21 ± 0.02 mg/kg, followed by sunflower honey from Iași, with a concentration of 0.14 ± 0.02 mg/kg. The elevated concentrations of Li, Sr, Ni, and Al in the honey samples from Tulcea, Botoșani, Iași, and Sibiu regions may be indicative of atmospheric pollution. Factors such as industrial emissions, agricultural practices, and human activities in these areas could contribute to the observed high levels of these elements in honey. Further investigation into the sources of pollution in these specific regions is warranted. The results obtained in the case of Li, Se, Ni, and Al concentrations are comparable to those presented in previous research, both at the national [32,33,34,35,36,37,38] and international level [39,40,41,42,43,44]. Heavy metal concentrations in honey reflect the surrounding environment, varying across samples and locations due to factors like floral sources, environmental contamination, local conditions, seasonality, and production practices [28]. Notably, beryllium (Be), vanadium (V), cobalt (Co), gallium (Ga), selenium (Se), rubidium (Rb), telluride (Tl), silver (Ag), bismuth (Bi), indium (In), cesium (Cs), barium (Ba), arsenic (As), mercury (Hg), and uranium (U) were undetectable in all the honey samples analyzed.

The exploration of elemental concentrations in honey samples represents a crucial facet of food safety and environmental monitoring. This comprehensive analysis encompasses a spectrum of chemical elements, each with its unique significance. The first group, including beryllium (Be), vanadium (V), cobalt (Co), gallium (Ga), selenium (Se), rubidium (Rb), tellurium (Tl), silver (Ag), bismuth (Bi), indium (In), cesium (Cs), barium (Ba), arsenic (As), mercury (Hg), and uranium (U), showed non-detectable or minimal levels.

The delineation of variations in the concentrations of the analyzed minerals within the honey samples is methodically presented based on the subsequent categorization. This categorization is guided by a calcification scheme that encompasses discerning factors such as honey type, production region, and the chronological aspect of production years. The first group, characterized by elements ranging from beryllium (Be) to uranium (U), exhibits either negligible or trace amounts in honey. This group encompasses elements that may have limited interaction with honeybees or floral sources, leading to minimal incorporation into honey.

Be–V–Co–Ga–Se–Rb–Tl–Ag–Bi–In–Cs–Ba–As–Hg–U > Al–Sr–Ni–Sr > Li > Cd > Pb > Cr > Cu > Zn > Mn > Fe > Na > Mg > Ca > K

Moving to the second group, comprising aluminum (Al), strontium (Sr), nickel (Ni), and lithium (Li), exhibited higher concentrations compared to the first group. Among these, aluminum (Al) and strontium (Sr) stood out with notable values, suggesting a more significant contribution to honey composition. A descending order of element concentrations was observed, with cadmium (Cd), lead (Pb), chromium (Cr), copper (Cu), zinc (Zn), manganese (Mn), iron (Fe), sodium (Na), magnesium (Mg), calcium (Ca), and potassium (K) exhibiting progressively lower values. Notably, potassium (K) emerged as the most abundant element, highlighting its significant presence in honey. This ranking provides valuable insights into the mineral composition of honey, with elements like potassium (K) being major contributors, while others, particularly those in the first group, play a minor role in shaping the elemental profile of honey. Understanding these variations contributes to the comprehensive assessment of honey quality and its potential nutritional and environmental implications. Transitioning to the second group, including aluminum (Al), strontium (Sr), nickel (Ni), and lithium (Li), we observe varying concentrations, suggesting diverse geological and botanical influences on honey composition. Aluminum (Al) and strontium (Sr) emerge with noteworthy levels, indicative of their potential impact on honey’s elemental profile. Continuing this analysis, the subsequent groups showcase a descending order of concentrations, with potassium (K) being the most abundant element. This hierarchy sheds light on the relative prevalence of elements in honey, offering critical information for nutritional assessment and quality control. Understanding the distribution of elements in honey samples is essential for assessing their origin, potential environmental contamination, and overall quality. The intricate interplay between environmental factors, floral sources, and bees’ foraging behaviors contributes to the complex matrix of honey composition. This study aims to unravel these intricacies, providing a foundation for informed decision-making in the realms of food safety, environmental health, and consumer well-being. The targeted elements, spanning from beryllium (Be) to potassium (K), provide insights into the intricate composition of honey and offer valuable information regarding environmental factors, floral sources, and potential health implications.

### 3.2. Elemental Dispersion According to the Types of Honey

The mineral composition of honey varies depending on its floral source or type, reflecting the unique characteristics of the plants from which bees gather nectar. Honey is a valuable source of essential minerals, including potassium, calcium, magnesium, sodium, phosphorus, iron, manganese, and zinc. These minerals contribute not only to the nutritional value of honey but also to its distinct flavor and color. Different types of honey, such as acacia, clover, or manuka, exhibit specific mineral profiles based on the plants prevalent in their respective regions. Understanding the mineral composition of honey provides valuable insights into its quality, nutritional benefits, and potential applications in various industries.

This study presents a detailed exploration of the mineral compositions of diverse honey types, encompassing sunflower, chestnut, rape, lavender, honeydew, heather, acacia, linden, and multifloral honey. By analyzing specific elements such as potassium, magnesium, iron, sodium, and calcium, this research delves into the intricate variations influenced by factors like soil characteristics, agricultural practices, and geographical origin. The subsequent discussion highlights the unique mineral profiles of each honey type, offering valuable insights into the impacts of floral sources and regional distinctions. These findings contribute to a nuanced understanding of honey’s quality and nutritional attributes and the broader implications for diverse honey varieties:i.*Sunflower honey*: Sunflower honey exhibited a relatively low concentration of potassium (K): 45.35 ± 30.40 mg/kg. This finding suggests that sunflower honey has a distinct mineral composition compared to other honey types. This lower concentration might be due to factors such as soil characteristics and agricultural practices in the sunflower production region.ii.*Chestnut honey*: Chestnut honey, on the other hand, displayed the highest concentration of potassium (K): 2116.41 ± 183.26 mg/kg. This high K content contributes significantly to the overall mineral composition of chestnut honey. The specific values indicate considerable variability in chestnut honey, potentially influenced by the chestnut trees’ unique soil and nutrient requirements.iii.*Rape*: The honey derived from rape in Tecuci showed noteworthy concentrations of potassium (K), with the highest values recorded in autumn rape (1012.05 ± 20.89 mg/kg) and spring rape (1284.66 ± 52.33 mg/kg). These results underline the impact of geographical origin on the mineral content of honey, with variations attributed to seasonal and agricultural factors.iv.*Lavender honey*: Lavender honey from the Vaslui region demonstrated elevated concentrations of magnesium (Mg), namely, 84.45 ± 7.75 mg/kg, and iron (Fe), namely, 75.91 ± 5.98 mg/kg. These high values suggest that lavender honey has distinct mineral characteristics, potentially influenced by the specific soil and environmental conditions in the Vaslui region.v.*Honeydew:* Honeydew honey displayed the highest sodium (Na) concentrations among the samples, amounting to 69.42 ± 6.32 mg/kg. The mineral composition of honey is significantly influenced by its floral source, as evidenced by the elevated sodium content in honeydew honey compared to other honey varieties.vi.*Heather honey:* Heather honey exhibited elevated potassium (K) concentrations, showcasing the impact of floral source on this honey’s mineral profile. The results suggest that heather honey has a distinctive composition with higher potassium levels.vii.*Acacia Honey:* Acacia honey was discussed in relation to variations in potassium (K) concentrations across different regions. This highlights the influence of geographical factors, including soil characteristics and agricultural practices, on the mineral content of acacia honey.viii.*Linden Honey:* Linden honey was mentioned in the context of potassium (K) concentrations, with values reported in previous studies. This suggests that the mineral composition of linden honey can vary across different regions, with the specific concentration reflecting the unique characteristics of linden nectar.ix.*Multifloral honey:* Multifloral honey exhibited fluctuations in calcium (Ca) concentrations, indicating that even within the category of multifloral honey, there can be variations in mineral content. This highlights the need for detailed analyses to understand the specific mineral profile of multifloral honey.x.*Honey from various regions—geographical influence:* This study analyzed honey samples from various regions, including Botoșani, Râmnicu Vâlcea, Târgu Bujoru, Vaslui, Brăila, Satu Mare, Iași, Tulcea, Mehedinți, Sibiu, and Tecuci. The geographical origins of honey significantly impacted the concentrations of various metals, underscoring the diverse mineral profiles of honey from different regions. This variability necessitates region-specific analyses to accurately assess honey quality and potential health implications.

In summary, the detailed discussions provide insights into the specific mineral compositions of different types of honey, the factors influencing their variability across regions and floral sources, and the potential implications for honey quality and nutritional attributes.

### 3.3. Elemental Dispersion According to the Origin of Honey

Honey production areas exert a direct influence on mineral concentrations in honey, as revealed by this study. The geographical origin of honey significantly influences its elemental composition, as evidenced by the regional variations in potassium (K), calcium (Ca), magnesium (Mg), sodium (Na), iron (Fe), manganese (Mn), zinc (Zn), copper (Cu), chromium (Cr), lead (Pb), cadmium (Cd), lithium (Li), strontium (Sr), nickel (Ni), and aluminum (Al). These findings underscore the importance of considering geographical provenance when evaluating honey’s quality and potential health implications. The findings underscore the need to consider local factors such as soil composition, agricultural practices, and environmental conditions when regulating and ensuring the quality and safety of honey. In light of the acquired results, it is evident that the geographical origins of the honey samples exerted a notable influence, as delineated below.

i.*Potassium (K)*: The substantial variation in potassium (K) concentrations underscores the intricate relationship between honey composition and geographical factors. Honey from Tecuci, particularly that derived from rape, exhibited remarkably high potassium (K) values, emphasizing the regional influence on mineral content. The distribution of potassium (K) is intricately tied to honey’s geographical origin, involving soil characteristics and agricultural practices. This points to the significance of local environmental conditions in shaping the mineral profile of honey.ii.*Calcium (Ca)*: The diverse concentrations of calcium (Ca) across different honey types and regions highlight the nuanced nature of mineral content in honey. Multifloral honey from Botoșani and Râmnicu Vâlcea displayed extreme values, showcasing variability influenced by regional factors. Târgu Bujoru emerged with the highest concentration of calcium (Ca), while Râmnicu Vâlcea followed closely. This regional disparity manifests the impact of local conditions on the mineral makeup of honey.iii.*Magnesium (Mg), Sodium (Na), and Iron (Fe)*: The regional and varietal disparities in magnesium (Mg), sodium (Na), and iron (Fe) concentrations highlight the dynamic nature of honey composition. This variability necessitates a comprehensive understanding of the provenance and type of honey for conducting accurate quality assessments and determining potential health implications. Lavender honey samples from the Vaslui region exhibited the highest concentrations of magnesium (Mg) and iron (Fe), while honeydew stood out with elevated sodium (Na) values. Regional variations in mineral content, particularly the documented low values in Botoșani, Iași, and Râmnicu Vâlcea, underscore the regional specificity of honey composition. These variations are attributed to factors such as soil composition and floral sources, highlighting the importance of considering geographical provenance when evaluating honey quality and potential health implications.iv.*Manganese (Mn)*: The elevated manganese (Mn) concentrations in the honey samples from Botoșani, Brăila, and Arad underscore the regional variability in honey’s mineral composition. This regional specificity highlights the importance of considering geographical provenance when evaluating honey’s quality and potential health implications, as Mn content can influence honey’s sensory and nutritional properties. A notable proportion of samples exhibited manganese concentrations below the detection limit, indicating variations in honey composition even within the analyzed set.v.*Zinc (Zn), Copper (Cu), and Chromium (Cr):* The encroachment of the legally permitted maximum concentrations of zinc (Zn) and copper (Cu) in some samples raises concerns about the impact of environmental factors, agricultural practices, and regional differences on honey quality. Râmnicu Vâlcea stood out with the highest zinc concentrations, and several samples exhibited elevated copper values, emphasizing the need for detailed analyses at the regional level. Chromium (Cr) concentrations, though lower in magnitude, displayed regional disparities, with Sibiu, Iași, and Mehedinți showing higher values. This suggests localized influences on honey composition.vi.*Lead (Pb) and Cadmium (Cd):* The alarmingly high prevalence of lead (Pb) and cadmium (Cd) that exceeded the legal limits in honey samples from Brăila, Tulcea, Botoșani, Iași, and Mehedinți regions raises significant food safety concerns. These findings warrant immediate attention from regulatory bodies and apicultural stakeholders to implement effective mitigation strategies and safeguard public health. Regions with concentrations below the detection limits, such as Tecuci, Braila, Satu Mare, Sibiu, and Râmnicu Vâlcea, exhibit a more favorable profile, emphasizing the need for stringent monitoring and regulation.vii.*Lithium (Li), Strontium (Sr), Nichel (Ni), and Aluminum (Al):* The highest concentrations of lithium (Li), strontium (Sr), nickel (Ni), and aluminum (Al) in the honey samples from Tulcea, Botoșani, Iași, and Sibiu regions suggest potential atmospheric pollution in these areas. Further investigation into pollution sources in these specific regions is warranted to understand and address the factors contributing to elevated levels of these elements in honey.

The extensive analysis of various elemental concentrations in honey samples provides crucial insights into the intricate interplay between geographical factors, environmental conditions, and honey composition. Regional variations in mineral content highlight the need for targeted monitoring and regulation to ensure the safety and quality of honey across different regions. This study underscores the importance of understanding the specific influences of soil composition, agricultural practices, and environmental factors on honey composition to implement effective quality control measures.

### 3.4. Elemental Dispersion According to the Origins of Honey

Correlation analysis of the mineral elements in honey provides valuable insights into the interrelationships among these elements, shedding light on the complex dynamics of honey composition. Understanding these correlations is crucial for several reasons. Firstly, they offer clues about shared environmental influences, such as soil characteristics, agricultural practices, and floral sources, that contribute to the presence of specific minerals in honey. Secondly, correlated mineral concentrations can be indicative of common pathways of uptake or contamination, revealing potential sources affecting honey quality. Thirdly, correlations aid in predicting the behavior of certain elements based on the presence or absence of others, contributing to the development of a comprehensive understanding of honey mineral composition. This knowledge is instrumental for quality control measures, ensuring safety and adherence to regulatory standards in honey production.

i.*Na/K (R^2^ = 0.444 **): Explanation*: The moderate positive correlation between sodium (Na) and potassium (K) may be attributed to commonalities in environmental factors or plant sources, as both elements are influenced by soil composition.ii.*Mg/K (R^2^ = 0.309 **): Explanation:* The moderate positive correlation between magnesium (Mg) and potassium (K) suggests a shared influence from factors like soil composition and agricultural practices, impacting the concentrations of both minerals.iii.*Mg/Na (R^2^ = 0.289 **): Explanation:* The moderate positive correlation between magnesium (Mg) and sodium (Na) indicates a potential linkage in their uptake from the environment, possibly through similar floral or soil sources.iv.*Al/Na (R^2^ = 0.288 **): Explanation:* The moderate positive correlation between aluminum (Al) and sodium (Na) could be influenced by environmental factors, with both elements showing parallel variations in concentration.v.*Cd/Na (R^2^ = 0.377 **): Explanation:* The moderate positive correlation between cadmium (Cd) and sodium (Na) might be due to commonalities in the sources of contamination or shared pathways of uptake from the environment.vi.*Zn/Ca (R^2^ = 0.498 **): Explanation:* The strong positive correlation between zinc (Zn) and calcium (Ca) suggests a potential relationship influenced by soil characteristics or specific floral sources common to the regions.vii.*Cd/Li (R^2^ = 0.252 **): Explanation:* The weak positive correlation between cadmium (Cd) and lithium (Li) could be indicative of shared geological or anthropogenic sources, influencing their presence in honey.viii.*Cd/Cu (R^2^ = 0.282 **): Explanation:* The weak positive correlation between cadmium (Cd) and copper (Cu) may reflect similar environmental pathways or contamination sources affecting both elements.ix.Ni/Cr (R^2^ = 0.61 **): *Explanation:* The strong positive correlation between nickel (Ni) and chromium (Cr) suggests a substantial commonality in their environmental sources or geochemical influences on honey composition in the respective regions.x.Al/K (R^2^ = 0.232 *): *Explanation:* The positive correlation between Aluminum (Al) and Potassium (K) suggests a shared influence or common environmental factors affecting their presence in honey. This may be linked to soil composition or agricultural practices.xi.Li/Na (R^2^ = 0.248 *): *Explanation:* The positive correlation between Lithium (Li) and Sodium (Na) indicates a potential common source or similar uptake mechanisms. Environmental factors influencing Na concentrations may also impact Li levels in honey.xii.Cu/Al (R^2^ = 0.231 *): *Explanation:* The positive correlation between Copper (Cu) and Aluminum (Al) suggests a potential shared environmental influence or similar pathways of uptake. This could be related to soil composition or other regional factors.

The identified correlations among mineral elements in honey underscore the intricate relationships between different components. These associations hint at underlying factors, such as geographical origin, environmental conditions, and agricultural practices, influencing the elemental composition of honey. Notably, correlations like Na/K, Mg/K, Mg/Na, Al/Na, Cd/Na, Zn/Ca, Cd/Li, Cd/Cu, and Ni/Cr suggest complex interdependencies that warrant further investigation. Understanding these correlations provides a foundation for exploring the nuanced dynamics of honey composition and informs the development of targeted quality control measures. These findings contribute to the broader comprehension of the multifaceted factors shaping the mineral content of honey across various regions (Table 4).

These negative correlations suggest potential antagonistic relationships between certain mineral elements in honey:i.Pb/K (R^2^ = −0.273 **): *Explanation:* A negative correlation between lead (Pb) and potassium (K) may indicate that regions with higher potassium levels tend to have lower lead concentrations, possibly influenced by soil characteristics or agricultural practices.ii.Zn/Na (R^2^ = −0.311 **): *Explanation:* The negative correlation between zinc (Zn) and sodium (Na) suggests an inverse relationship, implying that areas with higher zinc concentrations may exhibit lower sodium levels, possibly reflecting variations in floral sources or environmental conditions.iii.Sr/Mg (R^2^ = −0.213 **): *Explanation:* The negative correlation between strontium (Sr) and magnesium (Mg) implies an opposing trend. Regions with elevated magnesium content may show reduced strontium levels, highlighting potential geological or environmental influences.iv.Cr/Mg (R^2^ = −0.203 **) and Cr/Zn (R^2^ = −0.273 **): *Explanation:* Negative correlations between chromium (Cr) and magnesium (Mg), as well as chromium (Cr) and zinc (Zn), suggest that higher magnesium or zinc levels may be associated with lower chromium concentrations, indicating potential interactions influenced by regional factors.v.Mn/Ca (R^2^ = −0.206 *) and Cd/Zn (R^2^ = −0.251 *): *Explanation:* Negative correlations between manganese (Mn) and calcium (Ca), as well as cadmium (Cd) and zinc (Zn), hint at potential antagonistic relationships. Regions with higher manganese or zinc levels might exhibit lower calcium or cadmium concentrations, respectively, emphasizing the complexity of mineral interactions in honey.

Iron (Fe) levels in the analyzed honey samples did not show significant correlations with other mineral elements, suggesting a lack of direct linear relationships between Fe and potassium (K), sodium (Na), magnesium (Mg), calcium (Ca), lithium (Li), aluminum (Al), zinc (Zn), copper (Cu), strontium (Sr), chromium (Cr), manganese (Mn), nickel (Ni), cadmium (Cd), or lead (Pb). The absence of significant correlations suggests that iron concentrations in honey may be influenced by independent factors or intricate interactions not captured in the linear relationships explored in this study. Further investigation into the specific determinants of iron content in honey, such as soil characteristics, floral sources, or processing methods, is warranted to facilitate a comprehensive understanding of its behavior within the context of mineral composition.

### 3.5. Hierarchical Clustering and Principal Component Analysis Unveiling Elemental Profiles in Honey

The grouping of specific regions into distinct clusters, such as Râmnicu Vâlcea, Iași, Teleorman, and Târgu Bujoru, forming the first group, and Vaslui, Galați, Satu Mare, Brăila, Botoșani, Tulcea, Sibiu, Tecuci, Mehedinți, and Arad, forming the second group, suggests that the mineral composition of honey is profoundly influenced by the geographical factors unique to each region (Figure 1).

The observed clustering likely reflects a combination of various environmental, geological, and agricultural practices characteristic of each group. The presence of specific metals in honey is intricately tied to the soil composition, floral sources, and regional environmental conditions, thus contributing to a distinctive metal fingerprint for honey originating from different areas. The separation of these regions into two groups underscores the complexity of the interplay between geographical features and honey composition.

The observed grouping of honey samples by mineral content might reflect shared geological formations, climatic conditions, or other environmental factors within each region. This suggests honey’s potential as a geochemical marker, capable of revealing the unique fingerprint of the geographic area where it is produced.

The dendrogram-based classification of honey samples into two major groups based on metal concentrations offers a nuanced understanding of the regional variations in honey composition. The observed clusters highlight the importance of considering geographical origin as a significant factor influencing the mineral content of honey. This information is not only valuable for academic and scientific purposes but also has practical implications for quality control and regulation within the honey industry. Further research and detailed investigations into the specific geological and environmental factors driving these regional differences would contribute to a more profound comprehension of the intricate relationship between honey composition and geographical origin.

The construction of the second dendrogram, elucidating the proximity and divergence among the analyzed metals, has unveiled intricate relationships and interdependencies among the elements. The identified groupings underscore the underlying connections that extend beyond individual metal concentrations, offering valuable insights into the shared influences and reciprocal effects within the honey samples.

In the first group, the association between aluminum (Al) and potassium (K) suggests a potential interplay influenced by common factors. Simultaneously, the connection observed between cadmium (Cd) and sodium (Na), as well as the linkage between copper (Cu) and iron (Fe), hints at shared pathways or environmental influences affecting these metal pairs. The hierarchical structure also reveals the subordinate roles of Al and K, influenced by calcium (Ca), while Cd and Na are influenced by lithium (Li), and Mg exerts an influence on Cu and Fe (Figure 2).

Moving to the second group, the co-grouping of lead (Pb) and manganese (Mn), as well as nickel (Ni) and chromium (Cr) and strontium (Sr) and zinc (Zn), implies there are mutual influences within these metal pairs. These connections emphasize the intricate dynamics governing the concentrations of these metals, suggesting potential common sources or parallel uptake mechanisms.

The observed interconnections in both groups underscore the need for a holistic understanding of the factors influencing metal concentrations in honey. The dependencies identified in the dendrogram provide a basis for further exploration, prompting inquiries into the specific environmental, geological, or anthropogenic factors that orchestrate these complex relationships.

In conclusion, the dendrogram-based grouping of metals in honey reveals a nuanced network of connections, highlighting the multifaceted nature of mineral composition. This insight contributes to the broader comprehension of the intricate interplay of elements in honey and serves as a foundation for targeted investigations into the factors shaping metal concentrations in different regions.

The comprehensive analysis of the main components in honey, encompassing vital minerals and heavy metals such as K, Na, Mg, Ca, Li, Al, Fe, Cu, Zn, Sr, Cr, Mn, Ni, Cd, and Pb, has yielded noteworthy insights into the potential of these elements as discriminators for geographical origin and honey type. The differentiation achieved through these elements underscores their significance in characterizing the unique chemical profiles of honey samples, allowing for distinctions based on the specific environmental conditions, floral sources, and geological influences in diverse regions.

However, certain elements, namely, Fe, Ni, Li, Cd, Mg, and Se, exhibit a lower discriminatory capacity compared to their counterparts. The diminished ability of these elements to differentiate honey samples may be attributed to their ubiquity or shared occurrence across various geographical regions, minimizing their utility as exclusive markers for origin or honey-type determination. The nuances of these exceptions warrant further investigation into the specific factors contributing to their relatively limited discriminatory power.

In the context of principal component analysis (PCA), the incorporation of the first two factors (F1 and F2) has proven instrumental in capturing a substantial portion of the variance within the dataset. The cumulative contribution of F1 and F2, amounting to 71.52%, signifies the efficacy of these factors in summarizing the essential information encapsulated in the analyzed components. The significant loadings of these factors underscore their role in influencing the observed patterns and variations, providing a condensed yet representative depiction of the elemental composition of honey (Figure 3).

In conclusion, this study demonstrates the potential of the mineral and heavy metal analysis of honey samples to serve as a powerful tool for discriminating geographical origins and honey types. While certain elements exhibit reduced discriminatory capabilities, the overall success of the analysis, particularly with the aid of PCA, accentuates the value of these components in delineating the intricate chemical fingerprints of honey. The findings contribute to the broader understanding of honey composition and hold promise for applications in quality control, authentication, and traceability within the honey industry.

The integration of dendrograms depicting the proximity of honey samples based on metal analysis, along with the analysis of main components, provides a multifaceted understanding of the elemental composition of honey. The division of the dendrogram into distinct groups, reflective of geographical regions, underscores the regional specificity of metal concentrations in honey. This clustering aligns with the complex interplay of environmental factors, soil composition, and agricultural practices in shaping the elemental profiles of honey from different areas.

Simultaneously, the principal component analysis (PCA) emphasized the discriminatory power of essential elements, elucidating their roles in characterizing honey samples. Notably, exceptions such as Fe, Ni, Li, Cd, Mg, and Se, exhibiting lower discriminatory capabilities, warrant further investigation to unveil the nuanced factors influencing their presence in honey.

The statistical analysis of honey’s main components unravels a nuanced narrative of Romania’s diverse geological and environmental tapestry. The pronounced regional variations observed in the concentrations of potassium (K), calcium (Ca), magnesium (Mg), sodium (Na), iron (Fe), manganese (Mn), zinc (Zn), copper (Cu), chromium (Cr), lead (Pb), cadmium (Cd), lithium (Li), strontium (Sr), nickel (Ni), and aluminum (Al) in honey samples highlight the crucial role of local factors in honey analysis. Extremes in mineral concentrations, exemplified by Tecuci’s elevated potassium (K) levels in rape honey and rich magnesium (Mg) and iron (Fe) content of lavender honey in Vaslui, provide unique mineral profiles tied to specific regions and floral sources. However, environmental concerns arise as lead (Pb) and cadmium (Cd) concentrations in certain regions surpass legal limits, necessitating vigilant monitoring for potential contamination sources. Correlation patterns and complex interdependencies among minerals, such as the robust relationship between nickel (Ni) and chromium (Cr), deepen our understanding of honey composition and contamination pathways. Notably, iron (Fe) stands independently, hinting at unique influences not captured in linear relationships. In conclusion, this statistical exploration offers a comprehensive perspective on the intricate interplay of geological, environmental, and regional factors, advancing our understanding of honey quality and safety for use in future research and regulatory measures.

## 4. Conclusions

This comprehensive study embarked on a journey to unveil the intricate complexities of honey composition across Romania. By meticulously analyzing 61 samples from eight distinct regions, this research sheds light on the interplay between geography, botanical influences, and the resulting elemental makeup of honey. While certain elements remained undetectable or insignificant in quantity, this investigation yielded crucial insights with far-reaching implications for honey quality and safety. This analysis revealed a fascinating spectrum of elements in Romanian honey. Potassium (K) emerged as the undisputed king, showcasing significant regional variations. Essential minerals, particularly calcium (Ca), magnesium (Mg), and sodium (Na), exhibited a consistent presence across honey samples, while iron (Fe) also made notable contributions. Trace elements, including zinc (Zn), copper (Cu), and chromium (Cr), were detected in smaller quantities, with some regions exceeding established safety limits. This finding raises concerns about potential contamination pathways and underscores the need for stricter regulations.

The presence of lead (Pb) and cadmium (Cd) in levels exceeding legal boundaries in certain regions is a stark reminder of the potential threat posed by heavy metal contamination. Additionally, the elevated levels of lithium (Li), strontium (Sr), nickel (Ni), and aluminum (Al) point towards possible atmospheric pollution impacting honey composition. These findings necessitate further investigation to identify contamination sources and implement effective mitigation strategies.

This study delved further, exploring the connections between honey types and their elemental makeups. Sunflower honey exhibited the lowest K content, contrasting with the highest levels found in chestnut honey. Rape honey displayed intriguing seasonal and agricultural variations in its mineral composition, highlighting the influence of bee foraging behavior and agricultural practices. Lavender honey mirrored the unique fingerprint of its specific soil and environmental conditions, while honeydew honey showcased a distinct signature reflecting its floral source. Heather honey stood out with exceptionally high potassium levels, potentially linked to the specific plant species it interacts with. Acacia and Linden honeys served as prime examples of the influence of geographical origin on potassium concentrations. Finally, multifloral honey displayed fluctuations in calcium content, possibly reflecting the diverse floral sources utilized by bees.

The observed regional disparities in elemental composition underscore the profound influence of geography on honey. This reinforces the importance of considering local factors when assessing honey quality and safety. A one-size-fits-all approach for honey regulation is insufficient, and future efforts must prioritize region-specific monitoring and regulations. This study employed correlation analysis to unveil the intricate relationships between elements within honey samples. This analysis revealed interdependencies among elements, suggesting shared environmental influences. Interestingly, iron displayed no significant correlations, indicating a more independent behavior compared to other elements. Understanding these relationships is crucial for pinpointing potential contamination pathways and identifying the origins of specific elements found in honey.

This research also explored the potential of advanced techniques like dendrogram-based clustering and principal component analysis (PCA) for use in honey quality control, traceability, and authenticity assessment. The results demonstrate the effectiveness of metal analysis in these crucial areas. By applying these sophisticated tools, stakeholders can implement robust quality control measures and ensure consumers are protected from adulterated or contaminated honey products.

The intricate connection between honey composition and regional influences revealed in this study underscores the need for a multi-pronged approach. Firstly, region-specific monitoring and regulations are essential to ensure honey safety and quality. Secondly, honey producers and regulators can leverage the power of metal analysis techniques like PCA for quality control and traceability purposes. Finally, further research aimed at understanding the specific sources of potential contamination and exploring mitigation strategies remains crucial. This study serves as a steppingstone towards a more comprehensive understanding of Romanian honey, paving the way for robust quality control measures and informed regulations. By safeguarding the safety and authenticity of honey, stakeholders can ensure consumers continue to reap the health benefits and enjoy the delectable taste of this natural treasure.

## Figures and Tables

**Figure 1 foods-13-01253-f001:**
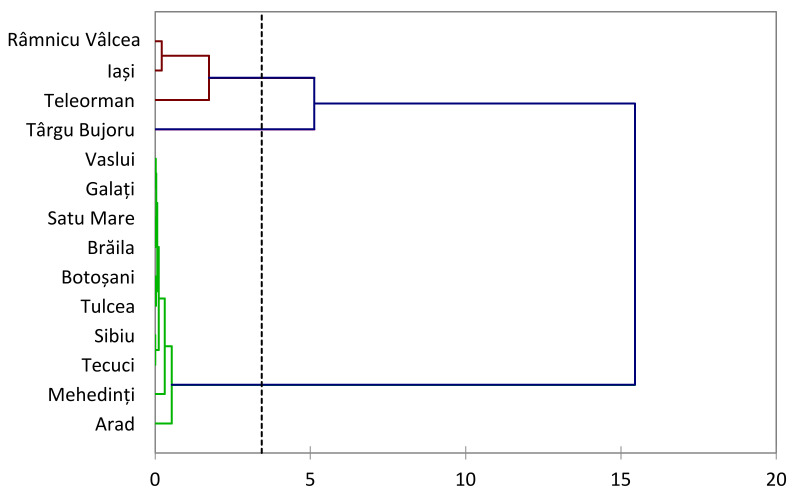
Geochemical clustering of regional honey compositions, providing insights into the influence of geographical factors on mineral profiles.

**Figure 2 foods-13-01253-f002:**
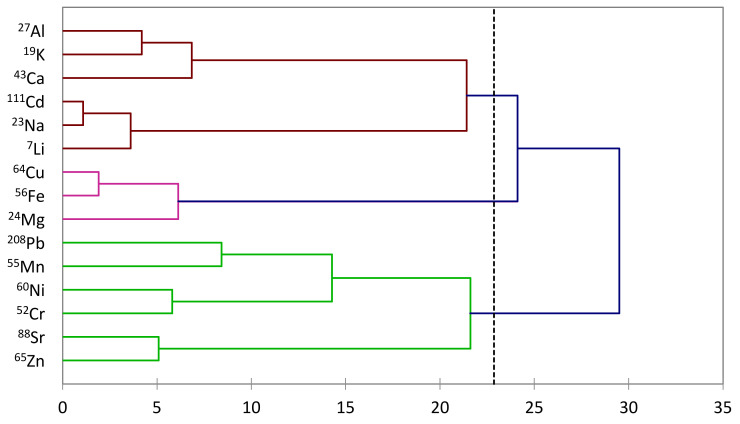
Dendrogram analysis unraveling the geographical impact on metal composition in honey, exploring interconnected dynamics and regions variances.

**Figure 3 foods-13-01253-f003:**
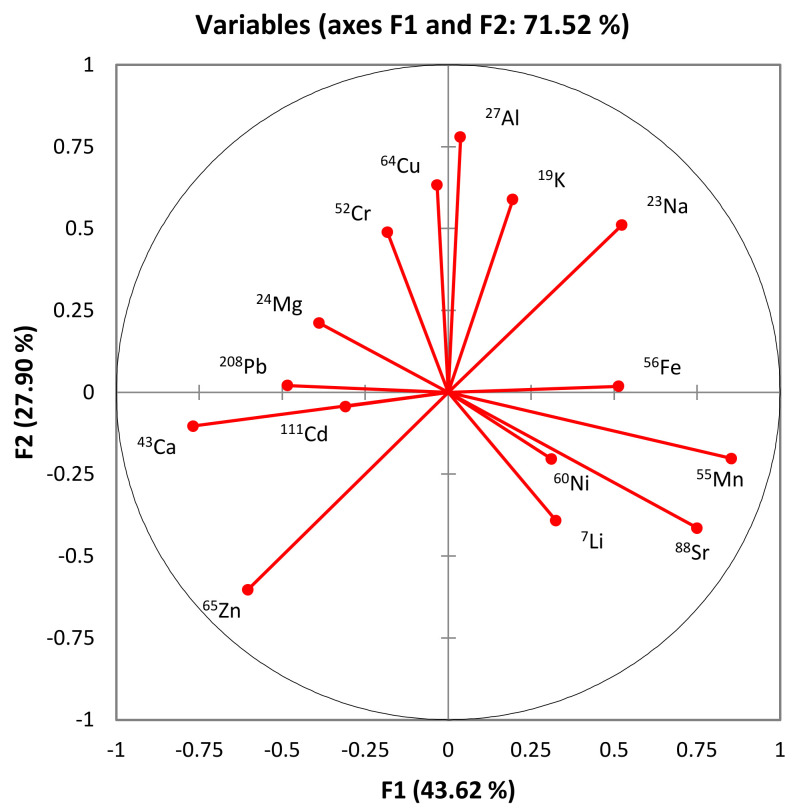
Insights into elemental signatures, unraveling the geochemical tapestry of honey composition and regional specificity.

**Table 1 foods-13-01253-t001:** Honey production details, maximum permissible levels (M.P.Ls), and elemental concentrations (mean ± SD) for floral honey collected from Romania (2018–2022) arranged by region and variety.

Seasons	Region (No Samples)	Honey Details	Area	^19^K (^mg/kg^) M.P.L.	^23^Na (^mg/kg^) M.P.L.	^24^Mg (^mg/kg^) M.P.L.	^43^Ca (^mg/kg^) M.P.L.	^7^Li (^mg/kg^) M.P.L.	^27^Al (^mg/kg^) M.P.L.	^56^Fe (^mg/kg^) M.P.L.	^64^Cu (^mg/kg^) M.P.L.	^65^Zn (^mg/kg^) M.P.L.	^88^Sr (^µg/kg^) M.P.L.
Maximum Permissible Levels (M.P.Ls)				–	–	–	–	–	–	–	0.50 mg/kg	1.00 mg/kg	–
2020	Southeast (2)	Multifloral	Galați	895.16 ± 43.74 ^ef^	19.12 ± 1.90 ^efghi^	32.37 ± 12.42 ^defg^	101.89 ± 9.29 ^f^	0.12 ± 0.03 ^fgh^	<LoQ ^e^	2.19 ± 0.34 ^hij^	0.64 ± 0.44 ^ef^	0.29 ± 0.17 ^hi^	<LoQ ^d^
2021	Southeast (3)	Linde		697.64 ± 51.41 ^ghi^	23.41 ± 41 ^ef^	27.13 ± 13.04 ^efghi^	80.95 ± 9.51 ^fgh^	0.16 ± 0.05 ^efgh^	0.01 ± 0.01 ^e^	61.65 ± 11.72 ^b^	0.17 ± 0.06 ^f^	0.16 ± 0.15 ^h^	<LoQ ^d^
2021	Southeast (1)	Acacia		275.34 ± 18.07 ^jk^	6.89 ± 2.06 ^î^	30.95 ± 2.90 ^efg^	26.77 ± 7.35 ^lmn^	0.06 ± 0.01 ^gh^	0.02 ± 0.03 ^e^	15.98 ± 6.57 ^efg^	0.17 ± 0.02 ^f^	<LoQ ^h^	<LoQ ^d^
2022	Southeast (2)	Sunflower	Târgu Bujoru	75.09 ± 6.08 ^l^	3.02 ± 0.71 ^î^	12.47 ± 0.72 ^îjk^	195.53 ± 16.45 ^cd^	0.33 ± 0.08 ^cdef^	<LoQ ^e^	6.21 ± 3.50 ^hij^	0.43 ± 0.23 ^f^	2.63 ± 0.49 ^dc^	<LoQ ^d^
2020	Southeast (2)	Spring rape	Tecuci	1284.66 ± 52.33 ^cd^	14.11 ± 5.69 ^fghiî^	13.69 ± 1.62 ^iîjk^	42.87 ± 6.48 ^jklm^	<LoQ ^h^	<LoQ ^e^	1.64 ± 0.58 ^ij^	<LoQ ^f^	0.63 ± 0.13 ^ghi^	0.10 ± 0.04 ^cd^
2020	Southeast (1)	Autumn rape		1012.05 ± 20.89 ^e^	3.11 ± 1.71 ^î^	20.83 ± 12.47 ^ghiîj^	31.37 ± 2.20 ^klmn^	<LoQ ^h^	<LoQ ^e^	3.55 ± 0.52 ^hij^	<LoQ ^f^	1.29 ± 0.25 ^fg^	0.17 ± 0.05 ^c^
2019	East (2)	Sunflower	Vaslui	630.96 ± 115.09 ^hiî^	9.85 ± 1.48 ^hiî^	44.88 ± 7.34 ^bcd^	78.67 ± 13.00 ^ghi^	0.28 ± 0.21 ^cdefg^	0.01 ± 0.01 ^e^	5.71 ± 1.70 ^hij^	1.83 ± 0.55 ^bcd^	3.23 ± 0.33 ^bc^	<LoQ ^d^
2018	East (2)	Linden		196.64 ± 8.42 ^kl^	11.86 ± 0.46 ^ghiî^	27.23 ± 9.52 ^efghi^	58.92 ± 14.98 ^iîj^	0.13 ± 0.02 ^fgh^	0.10 ± 0.07 ^cd^	11.50 ± 1.95 ^fgh^	0.86 ± 0.41 ^def^	<LoQ ^h^	<LoQ ^d^
2019	East (3)	Lavender		1004.16 ± 101.91 ^e^	25.66 ± 3.44 ^de^	84.45 ± 7.75 ^a^	27.81 ± 7.26 ^lmn^	0.06 ± 0.01 ^gh^	0.11 ± 0.04 ^c^	75.91 ± 5.98 ^a^	2.40 ± 1.25 ^b^	<LoQ ^h^	<LoQ ^d^
2021	Southeast (3)	Multifloral	Brăila	1842.76 ± 92.93 ^b^	35.46 ± 8.56 ^cd^	24.87 ± 5.54 ^fghiî^	151.44 ± 25.41 ^e^	0.62 ± 0.20 ^ab^	<LoQ ^e^	15.83 ± 2.88 ^efg^	0.79 ± 0.38 ^def^	2.18 ± 1.36 ^de^	<LoQ ^d^
2022	Southeast (1)	Linden		124.57 ± 5.95 ^l^	9.49 ± 2.60 ^iî^	24.87 ± 10.05 ^fghiî^	81.42 ± 6.65 ^fgh^	0.21 ± 0.15 ^defgh^	<LoQ ^e^	33.23 ± 9.52 ^c^	<LoQ ^f^	0.91 ± 0.69 ^ghi^	<LoQ ^d^
2021	Southeast (2)	Acacia + Linden		819.14 ± 210.14 ^fg^	11.09 ± 1.31 ^ghiî^	15.19 ± 5.74 ^hiîjk^	61.66 ± 14.63 ^hiîj^	<LoQ ^h^	<LoQ ^e^	29.02 ± 18.24 ^cd^	<LoQ ^f^	0.86 ± 0.63 ^ghi^	<LoQ ^d^
2020	Southeast (1)	Multifloral		1310.57 ± 115.01 ^cd^	40.07 ± 7.50 ^bc^	74.43 ± 14.85 ^a^	22.59 ± 3.80 ^mn^	<LoQ ^h^	<LoQ ^e^	20.19 ± 3.61 ^ef^	1.72 ± 1.84 ^bcde^	0.90 ± 0.06 ^ghi^	<LoQ ^d^
2021	Northwest (2)	Chestnut	Satu Mare	2116.41 ± 183.26 ^a^	33.96 ± 8.90 ^cd^	47.64 ± 6.16 ^bc^	185.25 ± 15.64 ^d^	<LoQ ^h^	0.21 ± 0.02 ^a^	<LoQ ^j^	<LoQ ^f^	<LoQ ^h^	0.02 ± 0.01 ^d^
2020	Northwest (1)	Acacia		83.40 ± 7.34 ^l^	20.09 ± 1.82 ^efghi^	11.62 ± 0.71 ^îjk^	18.50 ± 3.16 ^n^	<LoQ ^h^	0.07 ± 0.04 ^d^	<LoQ ^j^	2.01 ± 1.71 ^bc^	<LoQ ^h^	<LoQ ^d^
2020	Southeast (1)	Multifloral	Tulcea	318.77 ± 41.84 ^jk^	11.05 ± 1.97 ^ghiî^	18.52 ± 5.27 ^ghiîj^	48.66 ± 9.52 ^îjk^	0.47 ± 0.17 ^bc^	<LoQ ^e^	15.02 ± 5.17 ^fg^	0.97 ± 0.39 ^cdef^	<LoQ ^h^	<LoQ ^d^
2020	Southeast (2)	Sunflower		585.91 ± 33.69 ^iî^	42.53 ± 30.72 ^bc^	40.38 ± 0.42 ^bcde^	32.66 ± 5.98 ^klmn^	0.23 ± 0.15 ^defgh^	<LoQ ^e^	3.14 ± 2.31 ^hij^	4.31 ± 0.39 ^a^	<LoQ ^h^	<LoQ ^d^
2019	Southeast (1)	Linden		85.09 ± 15.20 ^l^	8.79 ± 4.73 ^iî^	35.78 ± 1.13 ^bcde^	51.22 ± 3.88 ^îjk^	0.75 ± 0.21 ^a^	<LoQ ^e^	2.47 ± 0.74 ^hij^	0.60 ± 0.10 ^f^	0.22 ± 0.25 ^h^	<LoQ ^d^
2021	Northeast (2)	Honeydew	Botoșani	1395.08 ± 77.99 ^c^	69.42 ± 6.34 ^a^	71.09 ± 8.22 ^a^	84.83 ± 13.41 ^fg^	0.69 ± 0.03 ^a^	<LoQ ^e^	3.09 ± 2.50 ^hij^	<LoQ ^f^	0.37 ± 0.18 ^hi^	<LoQ ^d^
2019	Northeast (3)	Multifloral		405.59 ± 16.04 ^j^	20.51 ± 3.10 ^efgh^	78.62 ± 6.09 ^a^	14.87 ± 3.45 ^n^	0.36 ± 0.17 ^cde^	<LoQ ^e^	16.23 ± 3.11 ^efg^	<LoQ ^f^	<LoQ ^h^	<LoQ ^d^
2019	Northeast (2)	Linden		76.17 ± 12.08 ^l^	8.61 ± 1.76 ^iî^	13.52 ± 1.03 ^iîjk^	17.69 ± 9.48 ^n^	0.11 ± 0.02 ^fgh^	<LoQ ^e^	<LoQ ^j^	<LoQ ^f^	0.18 ± 0.04 ^h^	<LoQ ^d^
2019	Northeast (1)	Sunflower		45.35 ± 30.40 ^l^	4.15 ± 0.64 ^î^	18.14 ± 4.48 ^ghiîj^	27.86 ± 10.71 ^lmn^	0.40 ± 0.23 ^cd^	<LoQ ^e^	3.61 ± 1.90 ^hij^	<LoQ ^f^	3.43 ± 0.27 ^b^	0.87 ± 0.27 ^a^
2021	Northeast (2)	Acacia + Linden	Iași	62.73 ± 9.26 ^l^	10.38 ± 1.13 ^ghiî^	28.38 ± 5.03 ^efgh^	222.31 ± 4.39 ^b^	<LoQ ^h^	<LoQ ^e^	<LoQ ^j^	0.69 ± 0.12 ^ef^	<LoQ ^h^	0.04 ± 0.01 ^d^
2020	Northeast (2)	Acacia		354.07 ± 59.45 ^j^	21.42 ± 2.83 ^efg^	2.17 ± 1.48 ^k^	33.52 ± 3.76 ^klmn^	0.81 ± 0.32 ^a^	0.02 ± 0.03 ^e^	<LoQ ^j^	1.00 ± 0.14 ^cdef^	<LoQ ^h^	0.03 ± 0.01 ^d^
2020	Northeast (2)	Sunflower		133.41 ± 70.41 ^l^	47.70 ± 13.21 ^b^	18.73 ± 4.10 ^ghiîj^	70.39 ± 3.36 ^ghiî^	0.39 ± 0.39 ^cd^	0.14 ± 0.02 ^b^	1.59 ± 0.40 ^ij^	2.03 ± 1.18 ^bc^	<LoQ ^h^	<LoQ ^d^
2021	Center (3)	Acacia	Sibiu	544.79 ± 24.04 ^î^	5.30 ± 2.98 ^î^	3.73 ± 0.19 ^k^	33.38 ± 5.03 ^klmn^	0.37 ± 0.15 ^cde^	<LoQ ^e^	32.05 ± 5.51 ^cd^	0.69 ± 0.36 ^ef^	0.36 ± 0.13 ^hi^	<LoQ ^d^
2020	Center (2)	Sunflower		1216.83 ± 103.83 ^d^	5.91 ± 2.55 ^î^	52.49 ± 15.17 ^b^	36.41 ± 11.80 ^klmn^	0.13 ± 0.03 ^fgh^	<LoQ ^e^	10.52 ± 2.47 ^ghi^	<LoQ ^f^	0.31 ± 0.23 ^hi^	<LoQ ^d^
2019	Southwest (1)	Sunflower	Râmnicu Vâlcea	623.97 ± 133.70 ^hiî^	3.68 ± 1.22 ^î^	48.50 ± 13.56 ^bc^	22.28 ± 6.08 ^mn^	<LoQ ^h^	<LoQ ^e^	5.46 ± 3.48 ^hij^	<LoQ ^f^	1.76 ± 1.27 ^ef^	<LoQ ^d^
2018	Southwest (1)	Multifloral		95.64 ± 23.64 ^l^	2.48 ± 1.48 ^î^	35.87 ± 5.59 ^cdef^	303.13 ± 19.57 ^a^	<LoQ ^h^	<LoQ ^e^	4.42 ± 4.86 ^hij^	<LoQ ^f^	6.46 ± 1.41 ^a^	<LoQ ^d^
2018	West (2)	Acacia	Arad	139.16 ± 44.70 ^l^	23.35 ± 3.32 ^ef^	7.92 ± 2.17 ^jk^	24.36 ± 7.72 ^mn^	0.47 ± 0.03 ^bc^	<LoQ ^e^	24.30 ± 11.87 ^ed^	0.10 ± 0.01 ^f^	0.34 ± 0.16 ^hi^	0.35 ± 0.13 ^b^
2020	South (3)	Sunflower	Teleorman	75.41 ± 9.19 ^l^	8.49 ± 5.52 ^iî^	45.51 ± 5.21 ^bcd^	64.03 ± 15.39 ^ghiîj^	0.46 ± 0.16 ^bc^	<LoQ ^e^	2.13 ± 0.37 ^hij^	<LoQ ^f^	2.45 ± 1.22 ^de^	<LoQ ^d^
2020	Southwest (2)	Acacia	Mehedinți	736.61 ±147.58 ^gh^	19.09 ± 0.78 ^efghi^	14.71 ± 3.78 ^hiîjk^	31.15 ± 3.03 ^klmn^	<LoQ ^h^	<LoQ ^e^	<LoQ ^j^	<LoQ ^f^	<LoQ ^h^	<LoQ ^d^
2022	Southwest (1)	Sunflower		305.38 ± 199.82 ^jk^	8.79 ± 0.69 ^iî^	73.42 ± 3.02 ^a^	213.28 ± 19.01 ^bc^	<LoQ ^h^	<LoQ ^e^	<LoQ ^j^	<LoQ ^f^	1.54 ± 1.30 ^fgh^	<LoQ ^d^
^1^ F				146.015	19.782	28.339	109.750	12.675	22.107	39.021	8.600	29.028	20.438
^2^ Sig.				***	***	***	***	***	***	***	***	***	***
Minimum–Maximum Values				45.35 -	2.48 -	2.17 -	14.87 -	<LoQ -	<LoQ -	<LoQ -	<LoQ -	<LoQ -	<LoQ -
				2116.41	69.42	84.45	303.13	0.81	0.21	75.91	4.31	6.46	0.87
AVERAGE				809.41 ± 577.92	20.76 ± 17.20	37.14 ± 22.39	77.49 ± 48.39	0.19 ± 0.20	0.03 ± 0.06	14.60 ± 20.72	0.55 ± 0.68	0.71 ± 0.68	0.05 ± 0.06

Average value ± standard deviation. ^1^ F = Fisher factor [Fisher’s method]. ^2^ Sig. = Significance. LoQ = limit of quantitation (lower than the limit of quantification). LoQ for ^19^K is 7.321 µ/L; LoQ for ^23^Na is 13.23 µ/L; LoQ for ^24^Mg is 9.00 µ/L; LoQ for ^43^Ca is 17.99 µ/L; LoQ for ^7^Li is 0.03 µ/L; LoQ for ^27^Al is 0.32 µ/L; LoQ for ^56^Fe is 17.57µ/L; LoQ for ^64^Cu is 0.14 µ/L; LoQ for ^65^Zn is 1.20 µ/L; LoQ for ^88^Sr is 0.48 µ/L. In the case of the following elements (^19^K, ^23^Na, ^24^Mg, ^43^Ca, ^7^Li, ^56^Fe, ^64^Cu, and ^65^Zn), the unit of measure is mg/kg, and in the case of (^88^Sr), the unit of measure is µg/kg. *** = shows that there are significant differences between the analyzed. variants.

**Table 2 foods-13-01253-t002:** Honey production details, maximum permissible levels (M.P.Ls,), and elemental concentrations (mean ± SD) regarding floral honey collected from Romania (2018–2022) according to region and variety.

Seasons	Region (No Samples)	Honey Details	Area	^9^Be (^µg/kg^) M.P.L.	^51^V(^µg/kg^) M.P.L.	^52^Cr (^mg/kg^) M.P.L.	^55^Mn (^mg/kg^) M.P.L.	^59^Co (^mg/kg^) M.P.L.	^60^Ni (^mg/kg^) M.P.L.	^70^Ga (^µg/kg^) M.P.L.	^79^Se (^µg/kg^) M.P.L.	^85^Rb (^µg/kg^) M.P.L.	^204^Tl (^µg/kg^) M.P.L.
Maximum Permissible Levels (M.P.Ls)				–	–	–	–	–	–	–			
2020	Southeast (2)	Multifloral	Galați	<LoQ	<LoQ	0.13 ± 0.03 ^ef^	1.12 ± 0.20 ^fgh^	<LoQ	0.13 ± 0.02 ^cde^	<LoQ	<LoQ	<LoQ	<LoQ
2021	Southeast (3)	Linde		<LoQ	<LoQ	0.13 ± 0.03 ^ef^	3.53 ± 1.06 ^cd^	<LoQ	0.07 ± 0.05 ^de^	<LoQ	<LoQ	<LoQ	<LoQ
2021	Southeast (1)	Acacia		<LoQ	<LoQ	0.14 ± 0.06 ^ef^	0.57 ± 0.25 ^h^	<LoQ	<LoQ ^e^	<LoQ	<LoQ	<LoQ	<LoQ
2022	Southeast (2)	Sunflower	Târgu Bujoru	<LoQ	<LoQ	0.45 ± 0.10 ^defg^	0.85 ± 0.09 ^gh^	<LoQ	0.08 ± 0.07 ^de^	<LoQ	<LoQ	<LoQ	<LoQ
2020	Southeast (2)	Spring rape	Tecuci	<LoQ	<LoQ	<LoQ ^g^	2.12 ± 0.41 ^ef^	<LoQ	<LoQ ^e^	<LoQ	<LoQ	<LoQ	<LoQ
2020	Southeast (1)	Autumn rape		<LoQ	<LoQ	<LoQ ^g^	0.89 ± 0.33 ^gh^	<LoQ	<LoQ ^e^	<LoQ	<LoQ	<LoQ	<LoQ
2019	East (2)	Sunflower	Vaslui	<LoQ	<LoQ	0.20 ± 0.04 ^efg^	<LoQ ^h^	<LoQ	0.18 ± 0.06 ^abcd^	<LoQ	<LoQ	<LoQ	<LoQ
2018	East (2)	Linden		<LoQ	<LoQ	1.74 ± 0.73 ^bc^	2.65 ± 1.01 ^de^	<LoQ	0.24 ± 0.09 ^abcd^	<LoQ	<LoQ	<LoQ	<LoQ
2019	East (3)	Lavender		<LoQ	<LoQ	0.67 ± 0.48 ^defg^	<LoQ ^h^	<LoQ	<LoQ ^e^	<LoQ	<LoQ	<LoQ	<LoQ
2021	Southeast (3)	Multifloral	Brăila	<LoQ	<LoQ	0.18 ± 0.04 ^efg^	2.33 ± 1.01 ^e^	<LoQ	<LoQ ^e^	<LoQ	<LoQ	<LoQ	<LoQ
2022	Southeast (1)	Linden		<LoQ	<LoQ	0.34 ± 0.18 ^defg^	5.42 ± 0.42 ^b^	<LoQ	<LoQ ^e^	<LoQ	<LoQ	<LoQ	<LoQ
2021	Southeast (2)	Acacia + Linden		<LoQ	<LoQ	0.13 ± 0.02 ^ef^	3.97 ± 1.24 ^c^	<LoQ	<LoQ ^e^	<LoQ	<LoQ	<LoQ	<LoQ
2020	Southeast (1)	Multifloral		<LoQ	<LoQ	0.78 ± 0.01 ^defg^	<LoQ ^h^	<LoQ	<LoQ ^e^	<LoQ	<LoQ	<LoQ	<LoQ
2021	Northwest (2)	Chestnut	Satu Mare	<LoQ	<LoQ	1.04 ± 0.42 ^cde^	<LoQ ^h^	<LoQ	0.07 ± 0.01 ^de^	<LoQ	<LoQ	<LoQ	<LoQ
2020	Northwest (1)	Acacia		<LoQ	<LoQ	0.89 ± 0.23 ^cdef^	2.54 ± 0.90 ^de^	<LoQ	<LoQ ^e^	<LoQ	<LoQ	<LoQ	<LoQ
2020	Southeast (1)	Multifloral	Tulcea	<LoQ	<LoQ	<LoQ ^g^	<LoQ ^h^	<LoQ	<LoQ ^e^	<LoQ	<LoQ	<LoQ	<LoQ
2020	Southeast (2)	Sunflower		<LoQ	<LoQ	<LoQ ^g^	<LoQ ^h^	<LoQ	<LoQ ^e^	<LoQ	<LoQ	<LoQ	<LoQ
2019	Southeast (1)	Linden		<LoQ	<LoQ	<LoQ ^g^	<LoQ ^h^	<LoQ	<LoQ ^e^	<LoQ	<LoQ	<LoQ	<LoQ
2021	Northeast (2)	Honeydew	Botoșani	<LoQ	<LoQ	0.14 ± 0.04 ^ef^	2.00 ± 0.47 ^efg^	<LoQ	<LoQ ^e^	<LoQ	<LoQ	<LoQ	<LoQ
2019	Northeast (3)	Multifloral		<LoQ	<LoQ	0.37 ± 0.15 ^defg^	0.89 ± 0.29 ^gh^	<LoQ	<LoQ ^e^	<LoQ	<LoQ	<LoQ	<LoQ
2019	Northeast (2)	Linden		<LoQ	<LoQ	<LoQ ^g^	7.17 ± 2.81 ^a^	<LoQ	<LoQ ^e^	<LoQ	<LoQ	<LoQ	<LoQ
2019	Northeast (1)	Sunflower		<LoQ	<LoQ	<LoQ ^g^	0.12 ± 0.11 ^h^	<LoQ	<LoQ ^e^	<LoQ	<LoQ	<LoQ	<LoQ
2021	Northeast (2)	Acacia + Linden	Iași	<LoQ	<LoQ	2.34 ± 0.71 ^b^	<LoQ ^h^	<LoQ	<LoQ ^e^	<LoQ	<LoQ	<LoQ	<LoQ
2020	Northeast (2)	Acacia		<LoQ	<LoQ	2.17 ± 0.17 ^b^	1.99 ± 0.20 ^efg^	<LoQ	0.29 ± 0.06 ^a^	<LoQ	<LoQ	<LoQ	<LoQ
2020	Northeast (2)	Sunflower		<LoQ	<LoQ	0.89 ± 0.23 ^defg^	<LoQ ^h^	<LoQ	0.09 ± 0.03 ^de^	<LoQ	<LoQ	<LoQ	<LoQ
2021	Center (3)	Acacia	Sibiu	<LoQ	<LoQ	3.29 ± 0.57 ^a^	0.63 ± 0.16 ^h^	<LoQ	0.30 ± 0.09 ^a^	<LoQ	<LoQ	<LoQ	<LoQ
2020	Center (2)	Sunflower		<LoQ	<LoQ	<LoQ ^g^	1.92 ± 0.50 ^efg^	<LoQ	0.22 ± 0.09 ^abc^	<LoQ	<LoQ	<LoQ	<LoQ
2019	Southwest (1)	Sunflower	Râmnicu Vâlcea	<LoQ	<LoQ	<LoQ ^g^	0.52 ± 0.08 ^h^	<LoQ	0.09 ± 0.04 ^de^	<LoQ	<LoQ	<LoQ	<LoQ
2018	Southwest (1)	Multifloral		<LoQ	<LoQ	<LoQ ^g^	<LoQ ^h^	<LoQ	<LoQ ^e^	<LoQ	<LoQ	<LoQ	<LoQ
2018	West (2)	Acacia	Arad	<LoQ	<LoQ	<LoQ ^g^	4.39 ± 1.34 ^bc^	<LoQ	0.26 ± 0.24 ^ab^	<LoQ	<LoQ	<LoQ	<LoQ
2020	South (3)	Sunflower	Teleorman	<LoQ	<LoQ	<LoQ ^g^	0.09 ± 0.04 ^h^	<LoQ	0.15 ± 0.03 ^bcd^	<LoQ	<LoQ	<LoQ	<LoQ
2020	Southwest (2)	Acacia	Mehedinți	<LoQ	<LoQ	3.66 ± 1.63 ^a^	<LoQ ^h^	<LoQ	0.15 ± 0.09 ^bcd^	<LoQ	<LoQ	<LoQ	<LoQ
2022	Southwest (1)	Sunflower		<LoQ	<LoQ	1.15 ± 0.18 ^cd^	<LoQ ^h^	<LoQ	0.18 ± 0.08 ^bcd^	<LoQ	<LoQ	<LoQ	<LoQ
F				-	-	13.810	23.345	-	6.183	-	-	-	-
Sig.				-	-	***	***	-	***	-	-	-	-
Minimum–Maximum Values				-	-	<LoQ	<LoQ	-	<LoQ	-	-	-	-
						3.66	7.17		0.30				
AVERAGE				-	-	0.54 ± 0.53	1.48 ± 1.31	-	0.05 ± 0.06	-	-	-	-

Roman letters denote significant differences (*p* ≤ 0.005) irrespective of the collection area and year. Commission Regulation (EC) No 1881/2006 dated 19 December 2006, establishing maximum levels for specific contaminants in food products. Off. J. Eur. Union 2006, L364/5–L364/24; Codex Alimentarius. Codex Alimentarius Standard for Honey 12–1981. Revised Codex Standard for Honey. Standards and Standard Methods (Vol. 11). 2001; Council Directive 2001/110/EC Regarding Honey. EU Off. J. 2002, L10, 47–52. BLD stands for Below the Detection Limit. (LoQ): LoQ for ^9^Be is 0.20 µ/L; LoQ for ^51^V is 4.04 µ/L; LoQ for ^52^Cr is 5.53 µ/L; LoQ for ^55^Mn is 0.039 µ/L; LoQ for ^60^Ni is 0.18 µ/L; LoQ for ^70^Ga is 0.04 µ/L; LoQ for ^79^Se is 0.03 µ/L; LoQ for ^85^Rb is 0.23 µ/L; LoQ for ^204^Tl is 0.2 µ/L. In the case of the following elements (^52^Cr, ^55^Mn, ^59^Co, and ^60^Ni), the unit of measure is mg/kg, and in the case of (^9^Be, ^51^V, ^70^Ga, ^79^Se, ^85^Rb and ^204^Tl), the unit of measure is µg/kg. Note: ^9^Be, ^51^V, ^59^Co, ^70^Ga, ^79^Se, ^85^Rb, and ^204^Tl were also analyzed but not detected in any sample. *** = shows that there are significant differences between the analyzed.

**Table 3 foods-13-01253-t003:** Honey production details, maximum permissible levels (M.P.Ls), and elemental concentrations (mean ± SD) in floral honey collected from Romania (2018–2022) according to region and variety.

Seasons	Region (No Samples)	Honey Details	Area	^208^Ag (^µg/kg^) M.P.L.	^209^Bi(^µg/kg^) M.P.L.	^115^In (^µg/kg^) M.P.L.	^133^Cs (^µg/kg^) M.P.L.	^137^Ba (^µg/kg^) M.P.L.	^75^As (^µg/kg^) M.P.L.	^111^Cd (^mg/kg^) M.P.L.	^201^Hg (^µg/kg^) M.P.L.	^208^Pb (^mg/kg^) M.P.L.	^238^U (^µg/kg^) M.P.L.
Maximum Permissible Levels (M.P.Ls)				–	–	–	–	–	–	0.02 mg/kg	–	0.20 mg/kg	–
2020	Southeast (2)	Multifloral	Galați	<LoQ	<LoQ	<LoQ	<LoQ	<LoQ	<LoQ	0.02 ± 0.01 ^b^	<LoQ	0.19 ± 0.04 ^bcde^	<LoQ
2021	Southeast (3)	Linde		<LoQ	<LoQ	<LoQ	<LoQ	<LoQ	<LoQ	0.01 ± 0.01 ^b^	<LoQ	0.21 ± 0.04 ^bcd^	<LoQ
2021	Southeast (1)	Acacia		<LoQ	<LoQ	<LoQ	<LoQ	<LoQ	<LoQ	<LoQ	<LoQ	0.10 ± 0.05 ^efg^	<LoQ
2022	Southeast (2)	Sunflower	Târgu Bujoru	<LoQ	<LoQ	<LoQ	<LoQ	<LoQ	<LoQ	0.08 ± 0.06 ^b^	<LoQ	0.12 ± 0.03 ^def^	<LoQ
2020	Southeast (2)	Spring rape	Tecuci	<LoQ	<LoQ	<LoQ	<LoQ	<LoQ	<LoQ	<LoQ	<LoQ	<LoQ	<LoQ
2020	Southeast (1)	Autumn rape		<LoQ	<LoQ	<LoQ	<LoQ	<LoQ	<LoQ	<LoQ	<LoQ	<LoQ	<LoQ
2019	East (2)	Sunflower	Vaslui	<LoQ	<LoQ	<LoQ	<LoQ	<LoQ	<LoQ	0.03 ± 0.02 ^b^	<LoQ	0.08 ± 0.02 ^fg^	<LoQ
2018	East (2)	Linden		<LoQ	<LoQ	<LoQ	<LoQ	<LoQ	<LoQ	0.01 ± 0.01 ^b^	<LoQ	<LoQ	<LoQ
2019	East (3)	Lavender		<LoQ	<LoQ	<LoQ	<LoQ	<LoQ	<LoQ	<LoQ	<LoQ	0.12 ± 0.03 ^def^	<LoQ
2021	Southeast (3)	Multifloral	Brăila	<LoQ	<LoQ	<LoQ	<LoQ	<LoQ	<LoQ	<LoQ	<LoQ	0.06 ± 0.05 ^fg^	<LoQ
2022	Southeast (1)	Linden		<LoQ	<LoQ	<LoQ	<LoQ	<LoQ	<LoQ	<LoQ	<LoQ	0.20 ± 0.17 ^bcde^	<LoQ
2021	Southeast (2)	Acacia + Linden		<LoQ	<LoQ	<LoQ	<LoQ	<LoQ	<LoQ	<LoQ	<LoQ	0.12 ± 0.04 ^def^	<LoQ
2020	Southeast (1)	Multifloral		<LoQ	<LoQ	<LoQ	<LoQ	<LoQ	<LoQ	<LoQ	<LoQ	0.20 ± 0.06 ^bcde^	<LoQ
2021	Northwest (2)	Chestnut	Satu Mare	<LoQ	<LoQ	<LoQ	<LoQ	<LoQ	<LoQ	0.04 ± 0.03 ^ab^	<LoQ	<LoQ	<LoQ
2020	Northwest (1)	Acacia		<LoQ	<LoQ	<LoQ	<LoQ	<LoQ	<LoQ	0.02 ± 0.01 ^ab^	<LoQ	<LoQ	<LoQ
2020	Southeast (1)	Multifloral	Tulcea	<LoQ	<LoQ	<LoQ	<LoQ	<LoQ	<LoQ	0.17 ± 0.04 ^a^	<LoQ	0.26 ± 0.13 ^abc^	<LoQ
2020	Southeast (2)	Sunflower		<LoQ	<LoQ	<LoQ	<LoQ	<LoQ	<LoQ	0.12 ± 0.01 ^a^	<LoQ	0.17 ± 0.02 ^bcdef^	<LoQ
2019	Southeast (1)	Linden		<LoQ	<LoQ	<LoQ	<LoQ	<LoQ	<LoQ	<LoQ	<LoQ	0.11 ± 0.01 ^defg^	<LoQ
2021	Northeast (2)	Honeydew	Botoșani	<LoQ	<LoQ	<LoQ	<LoQ	<LoQ	<LoQ	0.13 ± 0.14 ^a^	<LoQ	<LoQ	<LoQ
2019	Northeast (3)	Multifloral		<LoQ	<LoQ	<LoQ	<LoQ	<LoQ	<LoQ	0.05 ± 0.01 ^ab^	<LoQ	0.19 ± 0.02 ^bcde^	<LoQ
2019	Northeast (2)	Linden		<LoQ	<LoQ	<LoQ	<LoQ	<LoQ	<LoQ	0.01 ± 0.01 ^b^	<LoQ	0.31 ± 0.29 ^a^	<LoQ
2019	Northeast (1)	Sunflower		<LoQ	<LoQ	<LoQ	<LoQ	<LoQ	<LoQ	0.01 ± 0.01 ^b^	<LoQ	0.16 ± 0.03 ^cdef^	<LoQ
2021	Northeast (2)	Acacia + Linden	Iași	<LoQ	<LoQ	<LoQ	<LoQ	<LoQ	<LoQ	0.01 ± 0.01 ^b^	<LoQ	0.14 ± 0.03 ^def^	<LoQ
2020	Northeast (2)	Acacia		<LoQ	<LoQ	<LoQ	<LoQ	<LoQ	<LoQ	0.02 ± 0.01 ^b^	<LoQ	<LoQ	<LoQ
2020	Northeast (2)	Sunflower		<LoQ	<LoQ	<LoQ	<LoQ	<LoQ	<LoQ	0.04 ± 0.03 ^ab^	<LoQ	0.27 ± 0.05 ^ab^	<LoQ
2021	Center (3)	Acacia	Sibiu	<LoQ	<LoQ	<LoQ	<LoQ	<LoQ	<LoQ	<LoQ	<LoQ	<LoQ	<LoQ
2020	Center (2)	Sunflower		<LoQ	<LoQ	<LoQ	<LoQ	<LoQ	<LoQ	<LoQ	<LoQ	<LoQ	<LoQ
2019	Southwest (1)	Sunflower	Râmnicu Vâlcea	<LoQ	<LoQ	<LoQ	<LoQ	<LoQ	<LoQ	<LoQ	<LoQ	0.19 ± 0.04 ^bcde^	<LoQ
2018	Southwest (1)	Multifloral		<LoQ	<LoQ	<LoQ	<LoQ	<LoQ	<LoQ	<LoQ	<LoQ	0.14 ± 0.04 ^def^	<LoQ
2018	West (2)	Acacia	Arad	<LoQ	<LoQ	<LoQ	<LoQ	<LoQ	<LoQ	<LoQ	<LoQ	<LoQ	<LoQ
2020	South (3)	Sunflower	Teleorman	<LoQ	<LoQ	<LoQ	<LoQ	<LoQ	<LoQ	0.03 ± 0.01 ^ab^	<LoQ	<LoQ	<LoQ
2020	Southwest (2)	Acacia	Mehedinți	<LoQ	<LoQ	<LoQ	<LoQ	<LoQ	<LoQ	0.02 ± 0.01 ^b^	<LoQ	0.20 ± 0.05 ^bcde^	<LoQ
2022	Southwest (1)	Sunflower		<LoQ	<LoQ	<LoQ	<LoQ	<LoQ	<LoQ	<LoQ	<LoQ	0.21 ± 0.05 ^bcd^	<LoQ
F				-	-	-	-	-	-	7.380	-	9.202	-
Sig.				-	-	-	-	-	-	***	-	***	-
Minimum–Maximum Values				-	-	-	-	-	-	<LoQ -	-	<LoQ -	-
										0.17		0.31	
AVERAGE				-	-	-	-	-	-	0.03 ± 0.04	-	0.09 ± 0.07	-

Roman letters denote significant differences (*p* ≤ 0.005) irrespective of the collection area and year. Commission Regulation (EC) No 1881/2006 dated 19 December 2006, establishing maximum levels for specific contaminants in food products. Off. J. Eur. Union 2006, L364/5–L364/24; Codex Alimentarius. Codex Alimentarius Standard for Honey 12–1981. Revised Codex Standard for Honey. Standards and Standard Methods (Vol. 11). 2001; Council Directive 2001/110/EC Regarding Honey. EU Off. J. 2002, L10, 47–52. BLD stands for Below the Detection Limit (LoQ). LoQ for ^208^Ag is 0.17 µ/L; LoQ for ^209^Bi is 0.030 µ/L; LoQ for ^115^In is 0.011 µ/L; LoQ for ^133^Cs 0.021 µ/L; LoQ for ^137^Ba 0.17 µ/L; LoQ for ^75^As 0.74 µ/L; LoQ for ^75^Cd 0.07 µ/L; LoQ for ^201^Hg 0.20 µ/L; LoQ for ^208^Pb 0.20 µ/L; LoQ for ^238^U 0.08 µ/L. In the case of the following elements (^52^Cr, ^55^Mn, ^59^Co, and ^60^Ni), the unit of measure is mg/kg, and in the case of (^9^Be, ^51^V, ^70^Ga, and ^79^Se), the unit of measure is µg/kg. Note: ^208^Ag, ^209^Bi, ^115^In, ^133^Cs, and ^208^Ag were also analyzed but not detected in any sample. These elements, with limited presence, were excluded from detailed discussion due to their minor impact on honey composition. *** = shows that there are significant differences between the analyzed.

**Table 4 foods-13-01253-t004:** Elemental correlations (R^2^) for K, Na, Mg, Ca, Li, Al, Fe, Cu, Zn, Sr, Cr, Mn, Ni, Cd, and Pb in honey samples from various regions of Romania.

Elements	K	Na	Mg	Ca	Li	Al	Fe	Cu	Zn	Sr	Cr	Mn	Ni	Cd	Pb
**K**	1.000														
**Na**	0.444 **	1.000													
**Mg**	0.309 **	0.289 **	1.000												
**Ca**				1.000											
**Li**		0.248 *			1.000										
**Al**	0.232 *	0.288 **				1.000									
**Fe**							1.000								
**Cu**						0.231 *		1.000							
**Zn**		−0.311 **		0.498 **		−0.251 *			1.000						
**Sr**			−0.213 **							1.000					
**Cr**			−0.203 **						−0.273 **		1.000				
**Mn**			−0.345 **	−0.206 *				−0.230 *				1.000			
**Ni**											0.361 **		1.000		
**Cd**		0.377 **			0.252 **			0.282 **						1.000	
**Pb**	−0.273 **												−0.292 **		1.000

* Significant correlations were determined at the *p* < 0.05 level, indicating a 95% confidence level. ** Highly significant correlations were identified at the *p* < 0.01 level, signifying a 99% confidence level; the total sample size was N = 99. The correlation coefficient (R²) ranges from −1 to 1. A value of −1 indicates a perfect negative correlation, where one variable decreases as the other increases. Conversely, 1 indicates a perfect positive correlation, where both variables increase together. A value of 0 signifies no linear relationship. The closer the absolute value of R² is to 1, the stronger the correlation, whether positive or negative. These correlations provide insights into the intricate dynamics of mineral composition in honey, reflecting the complex interplay of environmental and regional factors. Further research is necessary to elucidate the specific mechanisms driving these relationships and their implications for honey quality.

## Data Availability

The original contributions presented in this study are included in the article/Appendix A; further inquiries can be directed to the corresponding author.

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
