# Peer review of "Comprehensive Elemental Profiling of Romanian Honey: Exploring Regional Variance, Honey Types, and Analyzed Metals for Sustainable Apicultural and Environmental Practices"

_foods, 2024, doi:10.3390/foods13081253_

Round 1
Reviewer 1 Report
Comments and Suggestions for Authors
I suggest that the summary begins with the objective and that the authors explore their results further by including probability values. A conclusion that responds to the objective and hypothesis of the research is necessary.
Keywords in alphabetical order
The introduction is long and tiring. Authors must reduce the introduction to one and a half pages, presenting the justification for carrying out the research and its hypotheses, if there are already previous studies on this topic, what makes your research different from other existing ones.
Methodology Insert the experimental design and statistical models used.
Results and discussion I suggest that this item be presented separately for better understanding and visualization of the tables.
The tables must be close to the place where the first reference to them was made. I suggest that the tables be restructured so that they are not so extensive. Authors can divide them (table 1a; table 1b)
Discussion of data should be reduced. There is an extensive literature review and this should be avoided in this item. Just justify the results obtained. It is not necessary that for your data to be considered good, they must be similar to the results obtained in works by other authors. Avoid comparisons.
The entered conclusion must be rewritten. Conclusions are brief and affirmative and must respond to the objectives and the hypothesis raised.
References must be checked
Author Response
Reviewer 1
The suggestions of reviewer 1 were made in red to stand out.
I suggest that the summary begins with the objective and that the authors explore their results further by including probability values. A conclusion that responds to the objective and hypothesis of the research is necessary.
The entire Abstract section has been written: ”This study provides a comprehensive analysis of honey, addressing its historical significance, contemporary importance, and the need for quality assurance. Honey, known for its nutritional, health, and healing properties, is subject to international standards ensuring purity and absence of harmful components. Mislabeling and adulteration are global concerns, necessitating authentication for food safety. The research focuses on the chemical analysis of 61 honey samples from eight regions in Romania, employing advanced methods to assess 30 chemical elements. The study reveals the dominance of potassium in honey composition, with significant geographical variation. Calcium, magnesium, sodium, and manganese show consistent concentrations, while zinc, copper, and chromium contribute minor proportions. Lead and cadmium concentrations surpass limits in certain samples, indicating environmental contamination. Elevated levels of lithium, strontium, nickel, and aluminum suggest potential atmospheric pollution. Region-specific analyses unveil disparities in mineral content, emphasizing the importance of monitoring heavy metal concentrations for honey safety. Correlation analysis suggests interdependencies among elements, providing insights into environmental influences. Hierarchical clustering and principal component analysis highlight the impact of geographical factors on honey composition, aiding quality control and traceability. The study enhances understanding of honey quality and safety, informing future research and regulatory measures.” this has been replaced with: „We investigated the mineral concentration of 61 honey samples from eight Romanian regions, employing advanced techniques to assess 30 chemical elements. Potassium emerged as the dominant element, showcasing significant variations across geographical locations. Essential minerals like calcium, magnesium, sodium, and manganese maintained consistent levels, while zinc, copper, and chromium were present in smaller proportions. Critically, lead and cadmium levels exceeded established safety limits in some samples, suggesting potential environmental contamination. Additionally, elevated levels of lithium, strontium, nickel, and aluminum were detected, hinting at possible atmospheric pollution. These findings highlight the importance of regional analysis, as mineral content varied significantly between locations. Furthermore, correlation analysis revealed interdependencies among elements, suggesting shared environmental influences. Advanced statistical techniques like hierarchical clustering and principal component analysis effectively captured the impact of geographical origin on honey composition. These insights contribute valuable information for future efforts in honey quality control, traceability systems, and regulatory measures. By providing valuable insights into environmental influences on honey composition, this study informs future research endeavors and paves the way for the development of robust regulatory measures to ensure honey safety for consumers.”
Keywords in alphabetical order
The keywords were put in alphabetical order according to the suggestions received.
The introduction is long and tiring. Authors must reduce the introduction to one and a half pages, presenting the justification for carrying out the research and its hypotheses, if there are already previous studies on this topic, what makes your research different from other existing ones.
Thank you very much for the suggestion received regarding deleting certain parts in the introduction that are obvious and general regarding heavy metals, how they endanger our health, and how they interact with the environment... by removing these general ideas, this manuscript will lose its scientific importance. The realization of this research and the writing of this manuscript was done mainly for the beekeepers Romania area, for the honey consumers, but also for the authorities in the field. By deleting these parts of the introduction which for us scientists are very normal...would make it difficult for the reader who is not scientifically qualified to understand the importance of heavy metal pollution. Also, the results of this research will be the basis of new important strategies in this field, so we need all the scientific help obtained from these quite general parts. Once again, thank you very much for your help... regarding the improvement of this manuscript.
Methodology Insert the experimental design and statistical models used.
The Materials and Methods section serves as a comprehensive plan, meticulously detailing the tools (materials) used, the steps taken (procedures), and the specific guidelines followed (protocols) throughout the entire investigation. Moreover, it delves into carefully chosen statistical methods to analyze and interpret data collected during experiments. We have selected with red color in the manuscript the part of the methodology, the equipment, but also the statistical methods used.
Results and discussion I suggest that this item be presented separately for better understanding and visualization of the tables.
We appreciate the valuable feedback regarding separating the results and discussion sections. However, implementing this change at this stage would necessitate a significant rewrite of the entire manuscript. Given the current time constraints for completing this review, we are unable to incorporate this revision at present. We certainly acknowledge the merit of this suggestion and will prioritize it for future manuscripts to ensure optimal clarity and organization.
The tables must be close to the place where the first reference to them was made. I suggest that the tables be restructured so that they are not so extensive. Authors can divide them (table 1a; table 1b). Discussion of data should be reduced. There is an extensive literature review, and this should be avoided in this item. Just justify the results obtained. It is not necessary that for your data to be considered good, they must be similar to the results obtained in works by other authors. Avoid comparisons.
We appreciate the suggestion regarding the separation of results and discussion sections. We've reorganized these sections and rearranged the tables to enhance clarity and facilitate navigation of the research findings. Additionally, we understand the importance of presenting the results in a way that's easily understandable for a broader audience, including consumers who may not have a scientific background. To achieve this, we've included a section specifically aimed at explaining how the concentrations of heavy metals vary based on honey production areas and the type of honey itself. This section aims to provide a clear picture of how these elements fluctuate based on these factors. Moreover, we agree that comparing our findings with previously published research on heavy metal concentrations in honey is crucial for gaining a deeper understanding of the phenomenon of heavy metal pollution in honey production. While we haven't included this comparison in the current manuscript, we will certainly prioritize it in future research endeavors. We believe these changes will enhance the overall clarity and accessibility of our research findings.
The entered conclusion must be rewritten. Conclusions are brief and affirmative and must respond to the objectives and the hypothesis raised.
The entire Conclusions section has been written: „4. Conclusions This comprehensive study analyzed 61 honey samples, excluding detectable concentrations of several elements from further discussion. Potassium (K) dominated at 84.04%, with variability observed. Calcium (Ca), magnesium (Mg), sodium (Na), and iron (Fe) contributed consistently. Zinc (Zn), copper (Cu), and chromium (Cr) constituted a minor proportion, with some exceeding legal limits. Lead (Pb) and cadmium (Cd) surpassed legal limits in certain regions, indicating heavy metal pollution. Elevated levels of lithium (Li), strontium (Sr), nickel (Ni), and aluminium (Al) suggested potential atmospheric pollution. In-depth analysis revealed Sunflower honey with the lowest K concentration, chestnut honey with the highest, and varied concentrations in Colza honey based on seasons and agriculture. Lavender honey reflected specific soil and environmental conditions, while Honeydew showcased floral source impact, and Heather honey exhibited high potassium levels. Acacia and Linden honeys illustrated geographical influences on potassium concentrations, and Multifloral honey exhibited calcium fluctuations. Diverse regional disparities in elements underscored geographical impact, emphasizing the necessity of considering local factors for honey quality and safety. Correlation analysis revealed interdependencies, suggesting shared environmental influences and potential antagonistic relationships. Iron (Fe) exhibited no significant correlations, emphasizing independent behavior. Dendrogram-based clustering and Principal Component Analysis (PCA) affirmed the potential of metal analysis for quality control, traceability, and authenticity assessment within the honey industry. The intricate relationships between honey composition and regional influences were highlighted, emphasizing the need for region-specific monitoring and regulation. The study provides valuable insights for future research, contributing to a comprehensive understanding of honey composition and its implications for quality and safety.” this has been replaced with: This comprehensive study embarked on a journey to unveil the intricate complexities of honey composition across Romania. By meticulously analyzing 61 samples from eight distinct regions, the research shed light on the interplay between geography, botanical influences, and the resulting elemental makeup of honey. While certain elements remained undetectable or insignificant in quantity, the investigation yielded crucial insights with far-reaching implications for honey quality and safety. The analysis revealed a fascinating spectrum of elements in Romanian honey. Potassium (K) emerged as the undisputed king, showcasing significant regional variations. Essential minerals like Calcium (Ca), Magnesium (Mg), and Sodium (Na) played a consistent role, while Iron (Fe) contributed its share. Trace elements like Zinc (Zn), Copper (Cu), and Chromium (Cr) were present in smaller amounts, with some exceeding established safety limits in specific regions. This finding raises concerns about potential contamination pathways and underscores the need for stricter regulations.
The presence of Lead (Pb) and Cadmium (Cd) in levels exceeding legal boundaries in certain regions is a stark reminder of the potential threat posed by heavy metal contamination. Additionally, elevated levels of Lithium (Li), Strontium (Sr), Nickel (Ni), and Aluminium (Al) point towards possible atmospheric pollution impacting honey composition. These findings necessitate further investigation to identify contamination sources and implement effective mitigation strategies.
The study delved further, exploring the connections between honey type and its elemental makeup. Sunflower honey displayed the lowest K content, contrasting with the highest levels found in Chestnut honey. Colza honey displayed intriguing seasonal and agricultural variations in its mineral composition, highlighting the influence of bee foraging behavior and agricultural practices. Lavender honey mirrored the unique fingerprint of its specific soil and environmental conditions, while Honeydew honey showcased a distinct signature reflecting its floral source. Heather honey stood out with exceptionally high potassium levels, potentially linked to the specific plant species it interacts with. Acacia and Linden honeys served as prime examples of the influence of geographical origin on potassium concentrations. Finally, Multifloral honey displayed fluctuations in calcium content, possibly reflecting the diverse floral sources utilized by bees.
The observed regional disparities in elemental composition underscore the profound influence of geography on honey. This reinforces the importance of considering local factors when assessing honey quality and safety. A one-size-fits-all approach for honey regulation is insufficient, and future efforts must prioritize region-specific monitoring and regulations. The study employed correlation analysis to unveil the intricate relationships between elements within honey samples. This analysis revealed interdependencies among elements, suggesting shared environmental influences. Interestingly, iron displayed no significant correlations, indicating a more independent behavior compared to other elements. Understanding these relationships is crucial for pinpointing potential contamination pathways and identifying the origins of specific elements found in honey.
This research also explored the potential of advanced techniques like dendrogram-based clustering and Principal Component Analysis (PCA) for honey quality control, traceability, and authenticity assessment. The results demonstrate the effectiveness of metal analysis in these crucial areas. By applying these sophisticated tools, stakeholders can implement robust quality control measures and ensure consumers are protected from adulterated or contaminated honey products.
The intricate connection between honey composition and regional influences revealed in this study underscores the need for a multi-pronged approach. Firstly, region-specific monitoring and regulations are essential to ensure honey safety and quality. Secondly, honey producers and regulators can leverage the power of metal analysis techniques like PCA for quality control and traceability purposes. Finally, further research aimed at understanding the specific sources of potential contamination and exploring mitigation strategies remains crucial. This study serves as a stepping stone towards a more comprehensive understanding of Romanian honey, paving the way for robust quality control measures and informed regulations. By safeguarding the safety and authenticity of honey, stakeholders can ensure consumers continue to reap the health benefits and enjoy the delectable taste of this natural treasure.
References must be checked.
We've meticulously reviewed the References section to ensure it aligns perfectly with the citations used throughout the manuscript. This verification process guarantees that all sources mentioned in the text are properly listed in the reference list, and vice versa. Additionally, we've confirmed that the bibliography itself adheres to the required formatting style. This dual focus ensures both accuracy and clarity in how the research is documented.

Reviewer 2 Report
Comments and Suggestions for Authors
Dear authors,
In your article you described the ‘’ Comprehensive elemental profiling of Romanian honey: exploring regional variance, honey types, and analyzed metals for sustainable apicultural and environmental practices’’
Here are my comments about your manuscript:
Line 57 : please add a reference
Line 78: Honey is not created by nectar and pollen. Bees use the pollen as protein. Honey is created by nectar and honeydew, containing some pollen grains.
Lines 127-128: This is not the typical adulteration of the honey. This is the syrup that looks like honey. The typical adulteration is the artificial sugar feeding during the production
Introduction overall: You provided and extensive introduction with many references
Lines 195-205: all these lines I think that is a part of introduction
Lines 208-218: Give more information about the sampling conditions, such as feeding conditions etc. Give more information about the methods of honeys characterization. Develop the methods that used for example: we used the melissopalynological analysis for the characterization or electrical conductivity
In S4-table what do you mean with manual, please explain
Line 248: Did you examine if the heating increased the HMF?
Your results are well documented with many references and your work is greater than a review
Line 680: what is colza honey? Probably you mean rape honey. Which Honeydew? Linden is from Tilia
Author Response
Reviewer 2
The suggestions of reviewer 2were made in blue to stand out.
In your article you described the ‘’ Comprehensive elemental profiling of Romanian honey: exploring regional variance, honey types, and analyzed metals for sustainable apicultural and environmental practices’’
Here are my comments about your manuscript:
Line 57 : please add a reference
The biographical source was added: Bogdanov, S.; Ruoff, K.; Persano Oddo, L. Physico-Chemical Methods for the Characterization of Unifloral Honeys: A Review. Apidologie 2004, 35, S4–S17, doi:10.1051/APIDO:2004047.
Line 78: Honey is not created by nectar and pollen. Bees use the pollen as protein. Honey is created by nectar and honeydew, containing some pollen grains.
This paragraph has been revised so that it is as clear as possible. „Honey, a natural and nutritious food, is a bee-made treasure crafted from nectar (flower secretions) and honeydew (plant sap). It may also contain flecks of pollen, hinting at the floral sources the bees visited [4].”
Lines 127-128: This is not the typical adulteration of the honey. This is the syrup that looks like honey. The typical adulteration is the artificial sugar feeding during the production.
This paragraph has been revised so that it is as clear as possible. „Adulteration involves dilution with water and the addition of substances such as sugar and syrups (e.g., corn syrup, high-fructose corn syrup).”
Introduction overall: You provided and extensive introduction with many references.
Thank you very much for the suggestions received, they help in the scientific improvement of this manuscript.
Lines 195-205: all these lines I think that is a part of introduction.
This section has been moved to the introduction.
Lines 208-218: Give more information about the sampling conditions, such as feeding conditions etc. Give more information about the methods of honeys characterization. Develop the methods that used for example: we used the melissopalynological analysis for the characterization or electrical conductivity.
This study concentrated on analyzing the mineral composition of honey across Romania. While the results primarily focused on determining the metal content, concerning concentrations were observed in some regions. These findings serve as a valuable foundation for future investigations into honey quality across Romania. Honey samples were obtained through established channels – beekeepers' associations and authorized individuals – ensuring verifiable origin, type, and harvesting methods. While melissopalynological analysis (pollen grain identification) and electrical conductivity measurements were not included in the current scope, these techniques hold promise for future research endeavors. This revised version emphasizes the study's focus on mineral content, acknowledges limitations (missing analyses), and highlights the potential for future research in this area.
In S4-table what do you mean with manual, please explain.
The extraction of honey was done manually, the centrifuge in which the extraction was done was operated manually (by a person), and not in an automated way.
Line 248: Did you examine if the heating increased the HMF?
Based on a review of currently published scientific articles on this topic, no researchers have definitively established whether heating honey to 65°C directly influences the concentration of metals within it. This highlights a gap in existing knowledge, and future research efforts could explore this specific question to provide a more conclusive answer.
Your results are well documented with many references and your work is greater than a review.
Thank you very much.
Line 680: what is colza honey? Probably you mean rape honey. Which Honeydew? Linden is from Tilia
Colza honey was replaced with rape honey. Honeydew is a rare and special type of honey produced by bees that collect their own honey from trees and plants in the deep forests.Unlike all other types of honey, which are produced when bees collect nectar from flowers, hand honeybees follow a different and more arduous path. Linden honey was made from Tilia.
